ecology, genetics, health and disease and epidemiology

*Escherichia coli*, wild animals, genetic diversity, pathogenicity, antimicrobial resistance, transmission

**Author for correspondence:**
Katherine M. Lagerstrom
e-mail: klager@stanford.edu

# The under-investigated wild side of *Escherichia coli*: genetic diversity, pathogenicity and antimicrobial resistance in wild animals

Katherine M. Lagerstrom[1] and Elizabeth A. Hadly[1,2,3]

[1]Department of Biology, [2]Stanford Woods Institute for the Environment, and [3]Center for Innovation in Global Health, Stanford University, Stanford, CA, USA

KML, 0000-0002-8747-0544

A striking paucity of information exists on *Escherichia coli* in wild animals despite evidence that they harbour pathogenic and antimicrobial-resistant *E. coli* in their gut microbiomes and may even serve as melting pots for novel genetic combinations potentially harmful to human health. Wild animals have been implicated as the source of pathogenic *E. coli* outbreaks in agricultural production, but a lack of knowledge surrounding the genetics of *E. coli* in wild animals complicates source tracking and thus contamination curtailment efforts. As human populations continue to expand and invade wild areas, the potential for harmful microorganisms to transfer between humans and wildlife increases. Here, we conducted a literature review of the small body of work on *E. coli* in wild animals. We highlight the geographic and host taxonomic coverage to date, and in each, identify significant gaps. We summarize the current understanding of *E. coli* in wild animals, including its genetic diversity, host and geographic distribution, and transmission pathways within and between wild animal and human populations. The knowledge gaps we identify call for greater research efforts to understand the existence of *E. coli* in wild animals, especially in light of the potentially strong implications for global public health.

## 1. Introduction

*Escherichia coli* residing in the gut microbiomes of wild animals has implications for human health in two major ways. First, *E. coli* can carry antimicrobial resistance (AMR) and virulence genes as well as share genetic information with its own and other species via horizontal gene transfer (HGT) [1]. Wild animals may therefore act not only as reservoirs of AMR and virulence genes that aid pathogenesis, but also as potential melting pots for novel gene combinations that could be more harmful to humans [2]. The presence of AMR genes in clinical cases of bacteriosis often leads to complications in treatment [3]. Second, efforts to identify the source of pathogenic *E. coli* in agricultural products are often complicated by the vast genetic diversity of *E. coli* and relatively narrow knowledge of its geographic distribution and host specificity. The understudied genetic diversity of *E. coli*, especially in its wild counterparts, along with its ability to gain, maintain and share AMR and virulence genes, call for greater scientific attention to protect global public health.

The impact of *E. coli* genetic diversity, virulence and AMR on biodiversity and conservation of wild animals is largely unknown. Stress, such as that resulting from habitat destruction, decreased food availability and other negative outcomes of anthropogenic environmental change, has been implicated as a factor related to shifts in wild animal microbiomes toward lower gut bacterial diversity and higher pathogen load [4,5]. This could lead to higher rates of

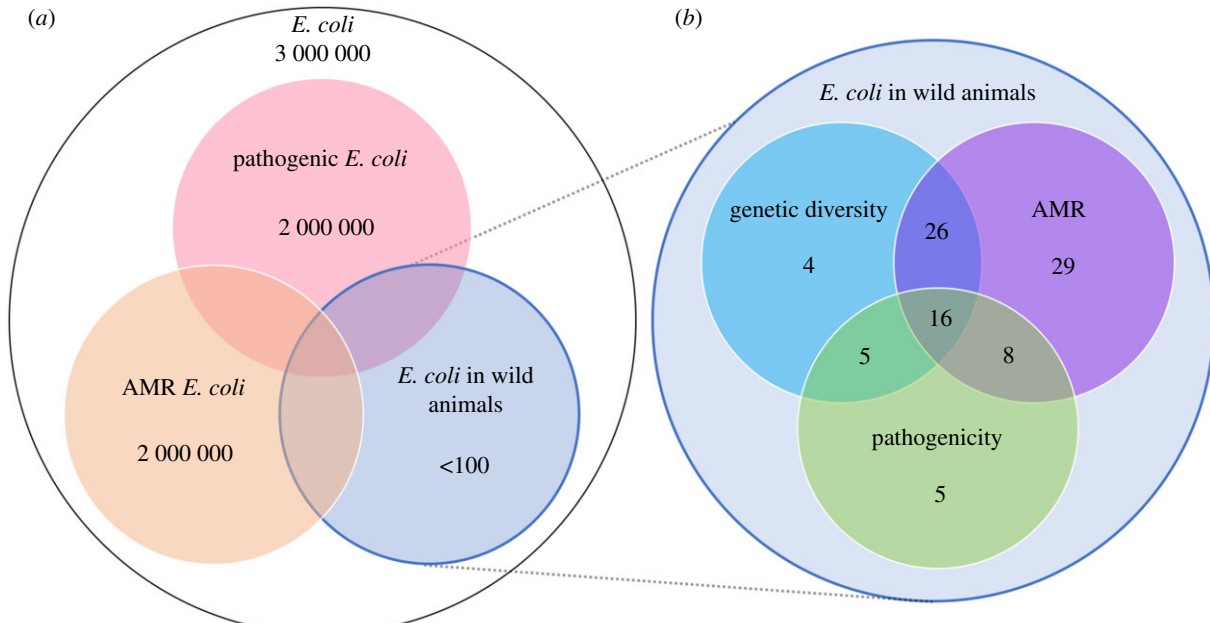

**Figure 1.** (*a*) Approximate number of *E. coli* studies falling into each category. (*b*) Breakdown of the 93 studies included in this literature review based on whether they assessed genetic diversity antimicrobial resistance or pathogenicity of *E. coli* in wild animal hosts. Electronic supplementary material, table S1 contains all references with their associated category assignments. (Online version in colour.)

pathogen and AMR shed into the environment and thus increase transmission among wild animals and potentially into human populations. It is still unclear whether or not harbouring pathogenic (to humans) strains, or subtypes, of *E. coli* causes an immune response in wild animals, but it has been shown to cause diarrhoea in cats, dogs and juvenile livestock [6,7]. It is also unknown if the presence of AMR or pathogenic strains of *E. coli* in wild animal microbiomes negatively impacts biodiversity and therefore conservation efforts.

This review seeks to assess the current understanding of *E. coli* in wild animals. Our literature review encompasses the body of work on *E. coli* in wild animal hosts, comprised of just 93 studies after explicitly excluding those considering domesticated animals, pets, livestock and captive zoo animals, which have been more studied for pathogenic *E. coli*. We identify three major gaps: (i) within the existing studies themselves (ii) in the geographic locations of studies to date and (iii) in the wild animal host species investigated. Addressing these gaps will prove essential for global human health and the preservation of many ecosystem services of high economic value [8–10].

## 2. Knowledge gaps in the literature

We reviewed the literature for studies on *E. coli* in wild animals, including all vertebrate hosts (birds, reptiles, fish and mammals) and excluding domestic and captive animals, livestock, humans and environmental *E. coli*. We excluded studies mentioning single *E. coli* strains common in foodborne illnesses because of their focus on human health and lack of contribution to the genetic understanding of *E. coli* in wild animals. We conducted a Google Scholar search in September, 2020. Searching for 'coli' 'E' OR 'Escherichia' anywhere in the article returned 3.51 million articles (figure 1). Including the term 'pathogenic' ('pathogenic' 'coli' 'E' OR 'Escherichia') resulted in 1.76 million articles and 2.02 million articles were returned by coli ('E' OR 'Escherichia') AND ('antimicrobial' OR 'antibiotic') AND ('resistant' OR 'resistance'). To target primary research articles and exclude those only citing previous studies in the references or text, the 'allintitle' search option was used. Searching for 'allintitle: coli wild E OR Escherichia -type' yielded 240 results. To target papers on *E. coli* in wild animals, we excluded the term 'type' to remove papers on wild-type *E. coli*. Entering 'allintitle: coli wild OR wildlife E OR Escherichia antimicrobial OR antibiotic OR resistant OR resistance -type' yielded 105 results. Excluding terms 'captive', 'domestic' and 'human' narrowed this to 77 articles. Entering 'allintitle: coli wild OR wildlife pathogenic OR EPEC OR STEC OR O157 -type' yielded 33 articles, but also included articles investigating *E. coli* in wild blueberries and wild thyme. The remaining unrelated articles were removed individually. To ensure our review was sufficient, a small assembly of articles was added idiosyncratically by combing the reference lists of those returned for any missed by our search terms. In total, 93 references on *E. coli* in wild animals were obtained and classified by whether they investigated AMR, pathogenicity, or genetic diversity (figure 1*b*). A reference was grouped within 'genetic diversity' if it assessed the genetic identity of *E. coli* beyond species identification, using such methods as multi-locus sequence typing (MLST), multiplex PCR [11], and in a few cases, whole-genome sequencing. Details of each reference classification are listed in electronic supplementary material, table S1.

Of over three million publications related to *E. coli*, only a handful to date address *E. coli* in wild animals. Fewer still investigate its genetic identity, as AMR is commonly the main focus (figure 1). Humans host over 30 genetically distinct resident *E. coli* and transiently host many more over their lifetime [12]. Frequently only one representative *E. coli* was taken from a single faecal sample per wild animal in these studies, a method which likely overlooks important within-host diversity and potentially under-represents the prevalence of AMR and pathogenic *E. coli* in wild animal populations. Furthermore, just over half of these papers genotyped the *E. coli* obtained. This oversite is stymying because it disallows phylogenetic

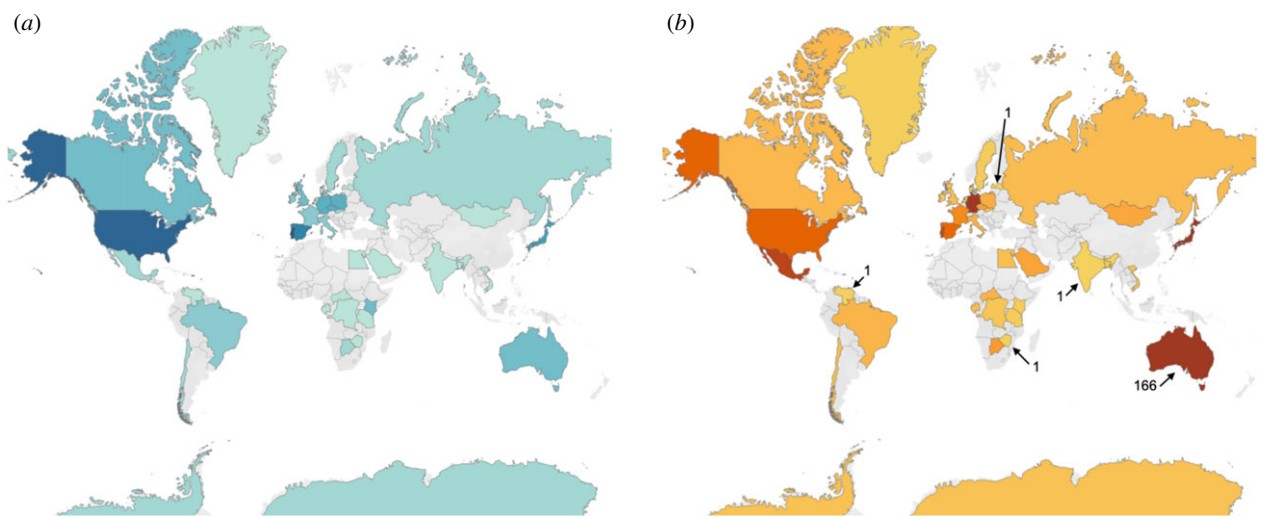

**Figure 2.** (*a*) Geographic gaps in research on *E. coli* in wild animals. Increasing gradient from pale blue, denoting a study count of 1, to dark blue, denoting a count of 10 (USA) or 11 (Portugal). No studies fitting our inclusion criteria were found in countries in grey. (*b*) Heat map depicting the number of wild animal host species investigated in each country. The colour gradient illustrates increasing number of species per country from yellow to dark orange. Min (1) and max (166) are labelled. electronic supplementary material, table S2 lists the number of studies and host species investigated in each country. (Online version in colour.)

studies that could provide important information on the level of genetic diversity and HGT within wild animal-hosted *E. coli*. This information may aid in tracking transmission pathways via analyses of genetic relatedness and modelling of mutation rates. Another oversite lies in the number of individual hosts sampled within each wild animal species, as many studies report on sample sizes less than 10 and often only one. Advancing research on the zoonotic disease potential of *E. coli* relies on the comprehensive assessment of its genetic diversity in wild animal populations.

The literature review revealed that *E. coli* has only been studied in wild animals in 40 countries and Antarctica (figure 2*a*). In just under half of these countries, there exists only one published study. Geographic biases in wild animal-hosted *E. coli* studies disallow addressing questions important to public health such as is the distribution of *E. coli* geographically determined, host specific or ubiquitous? Are there higher rates of AMR in countries with weaker controls on antibiotic use in the medical and agricultural sectors? Are rates of clinical cases of AMR pathogenic *E. coli* higher in countries where people have more contact with wild animals? Further disabling addressing these questions, *E. coli* has been investigated in just over 500 wild animal host species (figure 2*b*). This represents an inconsequential fraction of extant species, 79% of which have only been sampled once in a single geographic location (electronic supplementary material, table S3). A deeper sampling of wild animal-hosted *E. coli* across diverse species and geographic ranges will advance our understanding of transmission routes and commonness of spill-over between humans and wildlife, ultimately aiding in protecting global public health.

## 3. Summary of what is known

### (a) Genetic diversity of *Escherichia coli* in wild animals

*Escherichia coli* is one of the most genetically and phenotypically diverse microbial species known. Only about 6% of the pan genome (total number of genes in the species) is shared among all strains of *E. coli* [13,14]. The remaining greater than 90% is composed of variable 'accessory genes'.

Together with its capacity for homologous recombination (HR), or the exchange of genetic information between similar molecules of DNA, and HGT, the total gene pool of *E. coli* is essentially infinite [15]. In fact, the frequency of HGT in *E. coli* is so great that the evolution of its core genome (genes shared by all *E. coli*) is driven almost entirely by recombination, and therefore no single consensus phylogenetic tree exists for *E. coli* [16]. These features significantly complicate phylogenetic studies of *E. coli*.

To date, eight phylogenetic groups of *E. coli* have been defined: A, B1, B2, C, D, E, F and most recently, G [11,17], along with additional groups dubbed *Escherichia* cryptic clades [18]. The first study to identify *E. coli* in wild animal faecal samples and examine its genetic make-up found higher genetic diversity (measured by the Shannon diversity index; H, an index commonly used to characterize species diversity in a community) than that of most bacteria discovered to date [19]. Four years later, *E. coli* isolated from over 2300 non-domesticated vertebrates in Australia, including mammals, birds, fish and reptiles, showed similarly high phylogroup diversity, classified to one of four phylogroups defined at the time (15% A; 33% B1; 35% B2; 17% D) [20]. Over a decade later and following the acknowledgement of additional phylogroups, the genome sequences of 2244 *E. coli* isolates were obtained from a broad range of sources (mammalian, human and environmental) and similarly high phylogroup diversity was found (23% A; 47% B1; 13% B2; 6% D; 9% E; 2% F (prior to the definition of group G)) [21]. Distinct ecological roles have been suggested for each group. Groups A, B2, and D are more common in humans, though this distribution depends on the human population under investigation [22], while B1 tends to be common in animals and abiotic environments [23,24]. Groups B2 and D are most often commensal, but are more likely to carry virulence factors than A or B1 [25]. It is important to note that these ecological role classifications were made based on the now outdated method capable of assigning *E. coli* to one of four major phylogroups (A, B1, B2 or D). Further studies on the phylogroup identities of *E. coli* in wild animals will elucidate potential roles of the more recently defined phylogroups and allow enquiries regarding whether a species' phylogroup

**Figure 3.** The x-axis represents gradients of (*a*) increasing human population density (*b*) decreasing wildlife diversity (*c*) level of human impact from low (natural environments) to high (farming/livestock operations) and the *y*-axis indicates the proportion of resistant to commensal *E. coli* in an individual host's gut. The line illustrates the hypothesized increase in prevalence of AMR *E. coli* in host gut microbiomes corresponding to moving up each gradient. (Online version in colour.)

distribution is impacted by geography, latitude, host phylogeny or level of contact with humans and their domesticates.

## (b) AMR *Escherichia coli* in wild animals

Extensive use and misuse of antibiotics in human and veterinary medicine, as well as in agriculture and animal husbandry, has placed strong selective pressure on bacteria to evolve or acquire (via HGT) genes encoding AMR. In fact, the primary cause of the spread of AMR throughout the biosphere is human use of antibiotics [26]. Pathogenic *E. coli* is the most common cause of human urinary tract and bloodstream infections worldwide [27]. *Escherichia coli* causing these infections can carry AMR [28]. Combating AMR infections cost the US two billion dollars a year (Dall, 2018), but even this colossal effort is not enough, as each year, more than 23 000 deaths from AMR infections occur in the US alone (Fisher, 2017). The 2014 O'Neill Report estimated 700 000 deaths worldwide in 2014 were caused by one of six AMR species (including *E. coli*) and predicted that this number could reach 10 million by 2050 without successful intervention [29]. AMR has thus been identified as one of the largest threats to global health, food security and development by the World Health Organization [30].

Among the most commonly prescribed, beta-lactams are a broad class of antibiotics including penicillins and cephalosporins. Bacteria have developed the ability to produce an enzyme capable of inactivating beta-lactam antibiotics called beta-lactamase. Extended-spectrum beta-lactamase-producing *E. coli* (ESBL-*E. coli*) is a rising public health concern globally due to their ability to increase in prevalence rapidly via HGT and to confer multiple drug resistances. A recent review reported an eightfold increase in the intestinal carriage rate of ESBL-*E. coli* in humans over the past two decades alone [31]. A medical challenge for humans and domestic animals since the 80s, ESBL-*E. coli* was not discovered in wildlife until 2006 when Costa *et al.* surveyed birds of prey in Portugal, 36% of which harboured ESBL-*E. coli*. A series of supporting studies followed and have previously been summarized [2]. ESBL-*E. coli* have been isolated from wild birds on all continents except Australia (which has ESBL-*E. coli*-carrying bats, [32])

and Antarctica [33]. Thus, ESBL-*E.coli* is considered another form of environmental pollution [2].

Wild animals risk exposure to antimicrobial compounds and AMR bacteria via contact with anthropogenic sources such as human waste (garbage and sewage) and contaminated waters [34–36], livestock operations [37–39] or predation of affected prey, including livestock carcasses [40,41]. Evidence thus suggests a positive correlation between the level of human impact on an environment and the prevalence of AMR in wild animals (figure 3). The level of human impact is measured in part by population density, reductions in wildlife biodiversity and agricultural intensity, where areas of higher human population density and more intensive agricultural production often correspond to higher levels of AMR in proximate wild animals. Rolland *et al.* [42] compared the prevalence of AMR in groups of wild baboons (*Papio cynocephalus*) in Amboseli National Park, Kenya, finding low prevalence in those with little to no contact with humans, while 94.1% of samples from those close to a tourist lodge and its associated human waste carried multiple resistances. Livestock have been linked to a higher prevalence of AMR in proximate wild animals, including migratory birds and small mammals in several studies [37,38,43–47]. AMR *E. coli* prevalence was very high (71.43%) in Egyptian vultures (*Neophron percnopterus*) wintering at a livestock carcass dump in India [40]. Waterways have also been associated with higher AMR prevalence in wildlife [41,48,49], livestock [50] and diarrheal disease in humans [51].

The prevalence of AMR in wildlife is high enough for wildlife to be considered environmental reservoirs and potential melting pots of AMR by many authors on the subject [52–55]. Despite many human-used antibiotics having a natural source [56], sometimes derived from genes with other functions at lower expression levels or conferring other resistances (i.e. to toxic metals), allowing some selection for and maintenance of these genes naturally, the types and prevalence of AMR in wild animals cannot be explained by this alone. In fact, genes conferring resistance to antibiotics are often lost from bacterial communities in environments void of antibiotic pressure. This phenomenon was partially explained in a recent study identifying such driving forces as increased cost (reduced growth rate) of carrying a resistance

gene and intrinsic properties of microbial communities which favour the susceptible phenotype, though caveats may exist for AMR genes carried on conjugative plasmids [57]. AMR genes achieve high prevalence in bacterial communities under selective pressure through HGT. Though a gene capable of conferring AMR may have had an alternate function in its original host (i.e. regulatory or metabolic activity), following HGT, its integration into the metabolic networks of a new host is so improbable that its only function could be to confer resistance [56]. The opportunity for anthropogenic-sourced and naturally occurring AMR to mix in wild animals further enhances the potential threat for the creation of novel resistances and increasingly pathogenic *E. coli* in wild animal hosts.

Wildlife thus may facilitate evolutionary novelty and re-infection of human populations, particularly as contact between humans and wild ecosystems increases. We have yet to document a pathogenic strain of *E. coli* evolving in wildlife and spilling back into humans or livestock with higher pathogenicity. However, current investigative approaches are impractical, making documenting such an event nearly impossible. Further explorations are necessary across diverse wildlife species and geographic locations to accurately estimate AMR prevalence and thus the likelihood of re-inoculating humans [2].

## (c) Pathogenic *Escherichia coli* in wild animals

A limited number of studies have investigated pathogenic *E. coli* in wild animals, despite evidence that they harbour it [58–62]. Human and animal pathogenic *E. coli* share a common genetic background [63], though little effort has been made to disentangle what this implies about its transmission into and out of wild animal populations. Human contraction of pathogenic *E. coli* from wild animal sources has occasionally been documented. An outbreak resulting in 15 illnesses and two deaths was traced back to Oregon-grown strawberries contaminated with wild deer faeces [61]. Eight children were sickened by pathogenic *E. coli* after exposure to wild Rocky Mountain elk (*Cervus elaphus nelsoni*) faeces on a soccer field in Colorado [60]. Where protocols are in place to respond to such outbreaks, they are inconsistent and challenging to implement with current tools and limited understanding of pathogenic *E. coli* reservoirs and transmission. The inability to trace spillover events to a source is a cause for great concern. This was demonstrated at the 2019 San Diego County Fair, where an outbreak of pathogenic *E. coli* resulted in one death and many illnesses culminating with inconclusive investigations of food stands, environmental and animal samples and a claim that animals in the petting zoo were 'likely' the source [64]. The inability to trace such outbreaks to a source points to an important gap in monitoring foodborne pathogen transmission.

Because no comprehensive and universal database of wild animal-hosted *E. coli* exists, tracking the source of contamination in food products is not yet tractable when wildlife is involved. Indeed, a recent review assessing the risk of enteric pathogen spillover from wild birds into human populations concluded that the reason this source of the disease has been undermined is due to the lack of data [65]. While tracking *E. coli* is difficult and expensive, it is necessary, as crop contamination with pathogenic *E. coli* costs millions of dollars from produce loss and treatment of illness and leads to numerous deaths worldwide annually. Wild animals are often implicated as sources of *E. coli*

contamination in produce, often with little or no supporting data [66,67]. Drastic measures have been taken on many farmlands in California, where farm owners have abandoned integrated conservation practices and exclude wildlife by trapping, poisoning, fencing and clearing land around farms in response to consumer pressure to ensure food safety [67]. Despite these environmentally harmful measures, the prevalence of pathogenic *E. coli* actually increased in many cases by more than an order of magnitude between 2007 and 2013, indicating that removal of semi-natural land cover may be ineffective [66,68]. Exclusion or elimination of native species on and surrounding farmland, such as bees and insectivorous birds and bats, via habitat destruction, trapping or killing, could potentially lead to incalculable losses of ecosystem services provided by these species. Destruction of native land cover may also shift biological communities towards dominance by more reservoir-likely hosts, counterintuitively exacerbating food safety risks [68]. For instance, wild animals living more proximately to human activity likely have greater exposure to AMR and human pathogens, insinuating that farming environments favouring these species may pose an elevated risk to public health.

Use of poorly or untreated water for irrigation, the survival of *E. coli* in soils exposed to low-temperature composting practices, and farmworkers themselves are all potential sources of contamination [69]. To find effective interventions, integrative research should strive to align the interests of farmers and consumers with conservation efforts and environmental awareness [10,66,67]. Greater work to classify *E. coli* from a broad range of wild animal species will improve our ability to track contamination sources. Gaining knowledge about wildlife-harboured *E. coli* could remove false blame from wild animals or enable directed contamination source tracking and intervention. This would ultimately benefit the environment, wildlife conservation efforts and agricultural production simultaneously.

## 4. Potential routes of transmission

The correlation between human impact level and AMR prevalence in wildlife and the challenge of sourcing pathogenic *E. coli* spillover events call for deeper investigations into *E. coli* transmission routes. How *E. coli* transmits between wild animal hosts and humans has been little studied, but existing studies indicate *E. coli* may be readily shared between hosts across a range of environments (figure 4). Here we discuss three key categories potentially influencing transmission.

## (a) At the livestock–wildlife interface

A significant source of AMR in wildlife is livestock operations, which commonly treat their animals with antibiotics, often to prevent the spread of disease in overcrowded and unsanitary living conditions, but also to increase the growth rate of livestock. It is estimated that over 80% of all antibiotic sales in the US go to livestock [70]. Wastewater from livestock operations serves as an environmental reservoir and location for the propagation of AMR genes to other clinically important pathogens [50]. A review article on AMR in *E. coli* from farm animals classifies *E. coli* as an emerging global threat due to the development of 'dramatically high levels of antibiotic resistance to multiple classes of drugs' [71]. Increasingly more studies reveal similarly high levels of AMR in *E. coli*

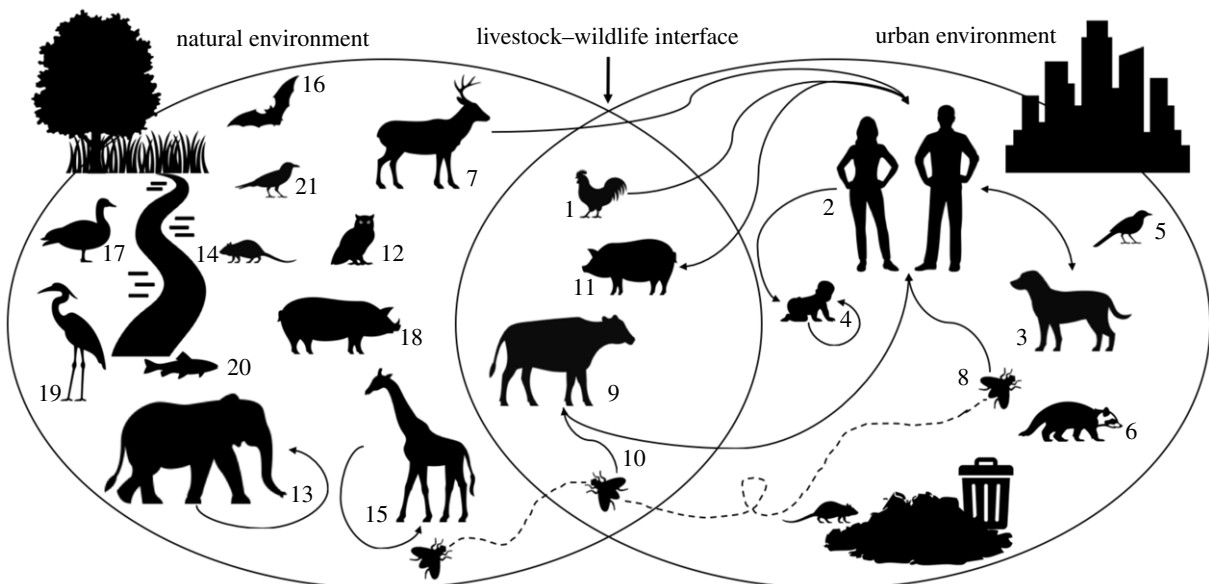

**Figure 4.** Diagram depicting where *E. coli* can reside in natural and urban environments. The intersection represents environments where captive, domesticated or livestock animals interact with both humans and wild animals. Arrows indicate confirmed pathways of *E. coli* transmission. Transmission, yet to be investigated, likely occurs between other species pictured. Numbers correspond to references on *E. coli* in the associated species listed in electronic supplementary material, table S4.

from both livestock and sympatric wildlife, indicating directional transmission [39,72,73]. Common pathogenic forms of *E. coli* were found in both wild birds and sympatric free-range cattle and wild geese in California, suggesting either a common environmental source of contamination or potential transmission between species [74]. AMR *E. coli* were significantly more prevalent in wild rodents in high livestock density areas than any other environment investigated by Guenther *et al.* in Germany, suggesting transmission from livestock to rodents [47]. Small mammal populations may thus serve as biological indicators of AMR pollution and *E. coli* transmission at local scales, importantly surrounding livestock facilities. Deeper investigations into the impacts of livestock operation wastewater and antibiotic use on sympatric wildlife will aid both food safety management and habitat conservation efforts.

### (b) Carried by birds

Another significant factor in *E. coli* transmission is the geographic movement of the host. For example, many wild bird species migrate long distances. Along the way, they excrete faeces, likely containing AMR and pathogenic *E. coli*. This is a substantial mechanism of spread, especially as birds more easily access remote areas, potentially impacting these ecosystems in yet unknown ways [2,75]. A comparison of migratory wild birds from Saxony-Anhalt, Germany to those in the secluded reaches of the Gobi-Desert, Mongolia found that overall rates of AMR *E. coli* were surprisingly similar, indicating that migration likely played a role in its dissemination [76]. AMR *E. coli* was also reported in remote populations of glaucous-winged gull (*Larus glaucescens*) on Commander Islands, Russia, where *E. coli* belonging to the globally disseminated human pathogen O25b-ST131 (an ESBL-*E. coli*) was first detected in a wild animal [77]. AMR *E. coli* has even been discovered in birds in as secluded environments as the Arctic [78]. Comparably high levels of AMR were detected in birds caught in both rural and urban areas in Michigan, implying that the movement of birds between rural and urban areas fuels the

spread of AMR [79]. This is especially concerning when wild birds come into contact with agricultural produce fields. Rivadeneira *et al.* found that wild birds carried pathogenic *E. coli* between CAFOs (concentrated animal feeding operations) and leafy green fields [80]. Wild birds also played a role in transmission among dairy farms in Ohio, where LeJeune *et al.* [81] identified pathogenic *E. coli* in cattle and sympatric starling populations that they tracked moving between farms [75]. Birds may thus serve as both widespread, even cross-continental biological indicators of AMR pollution and physical transmitters of AMR and pathogenic strains of *E. coli*.

### (c) Between individuals in wild populations

Wild animals likely first encounter *E. coli* via vertical transmission, as *E. coli* are among the first bacteria to colonize the gut microbiomes of human neonates [82]. Evidence suggests that subsequent encounters may partially be determined by social interactions. Giraffes closely connected in a co-occurrence network were more likely to have the same strains of *E. coli* than those rarely seen together, indicating that social contact networks may be able to predict *E. coli* sharing between individuals [83]. A study in wild elephants found that *E. coli* strains were more randomly distributed, with transmission patterns dominated by habitat and host traits, suggesting that social structure alone may not determine transmission, but that it may interact with exogenous factors such as the spatial distribution of waterholes and individual behaviour during drought [84]. Another study in the mountain brushtail possum (*Trichosurus vulpecula*) suggested that strain-sharing was better predicted by host contact than spatial proximity [85]. These inconclusive results demand greater research efforts into potential routes of transmission through wild animal populations.

### 5. Conclusion

Studies to date investigating the genetic diversity, distribution, pathogenicity and AMR of *E. coli* in wild animals

have provided a foundation for our understanding of the bacterial species' wild side. However, significant knowledge gaps remain regarding the potential for wild animals to act as reservoirs of AMR and pathogenic *E. coli*, as well as the likelihood and frequency of transmission between humans and wildlife. Preliminary studies show that wild animals host a diversity of *E. coli,* including AMR and pathogenic strains, and may act as vehicles of transmission and melting pots for the creation of new and potentially more dangerous strains that could threaten global health and food production. But because so few studies exist on *E. coli* in wild animals, the level of concern this public health threat ought to elicit remains difficult to assess.

Future research should prioritize investigations into the routes and mechanisms of transmission, especially at wildlife–livestock interfaces which offer significant opportunities for transmission of AMR and pathogenic *E. coli* between domestic and wild animal populations. Future work should also prioritize replicate sampling of individuals from a breadth of wild animal species across their geographic ranges to provide insight into the distribution and genetic diversity of *E. coli* both within an individual wild animal host as well as within and between wild animal populations. This will elucidate whether species that are significant reservoirs of *E. coli* differ from those that are of conservation concern, an important consideration for disease management and wildlife conservation. These research priorities could substantially improve current protocols for contamination source tracking and aid in the curtailment of future spillover into human populations. They will also allow us to better gauge the threat posed by wild animal-hosted *E. coli* to global public health.

Zoonotic disease emergence is a rising problem exacerbated by increasing human invasion of wild areas through continued urbanization and resource extraction, providing more opportunities for transmission between a wild animal and human populations [86]. We are already seeing signs of this spread, as game (kangaroo in Australia) and bushmeat (desert warthog (*Phacochoerus aethiopicus*), common duiker (*Sylvicapra grimmia*) and African buffalo (*Syncerus caffer*) in the Democratic Republic of the Congo) have been found contaminated with *E. coli* [87,88]. The increasing presence of AMR in human clinical cases of infection as well as in wild animal populations only further confounds this problem, as it will become increasingly challenging to combat these diseases as previous antibiotic treatments become ineffective. To combat as yet unknown challenges, we urgently need a better understanding of *E. coli* in wild animals to guide us in preventing spillover, tracking and curtailing contamination incidents, reducing the spread of AMR, and ultimately protecting global human health in the Anthropocene.

Data accessibility. This article has no additional data.

Authors' contributions. K.M.L. and E.A.H. conceived the topic and scope of this review article. K.M.L. conducted the literature review and drafted the manuscript. E.A.H. provided critical revisions to the manuscript. Both authors gave final approval for publication and agree to be held accountable for the work performed therein.

Competing interests. We declare we have no competing interests.

Funding. We received no funding for this study.

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
