## [Peer Review File · Proceedings of the Royal Society B: Biological Sciences]

Review History

RSPB-2020-2526.R0 (Original submission)

Review form: Reviewer 1

Recommendation

Reject – article is not of sufficient interest (we will consider a transfer to another journal)

Scientific importance: Is the manuscript an original and important contribution to its field?

Acceptable

General interest: Is the paper of sufficient general interest?

Good

Quality of the paper: Is the overall quality of the paper suitable?

Good

Is the length of the paper justified?

Yes

Should the paper be seen by a specialist statistical reviewer?

No

Do you have any concerns about statistical analyses in this paper? If so, please specify them explicitly in your report.

No

It is a condition of publication that authors make their supporting data, code and materials available - either as supplementary material or hosted in an external repository. Please rate, if applicable, the supporting data on the following criteria.

Is it accessible?

Yes

Is it clear?

Yes

Is it adequate?

Yes

Do you have any ethical concerns with this paper?

No

Comments to the Author

Manuscript RSPB-2020-2526 is an interesting and well-written review highlighting the sparse literature available to date on E. coli in wildlife, with an emphasis on antimicrobial resistance. I have mostly minor comments throughout. My primary concern with publication of the manuscript in Proceedings B is that many of the components are highlighted elsewhere in the literature. I believe it will make a nice contribution to the literature but isn't a major enough advancement for Proceedings B.

My primary big comment for the authors in revising the manuscript is that I think the review should have greater emphasis on livestock-wildlife transmission. For example, Fig. 3 focuses on an axis of increasing human population density and urban impact. However, exchange of E. coli (and other foodborne pathogens) often occurs at the wildlife-livestock interface rather than urban interface and can also be the source of AMR genes. Michele Jay-Russell's lab has quite a lot of literature on the matter, but to give some specific examples of a few refs of many that show or suggest wildlife-livestock transmission of foodborne pathogens: Carlson et al. (2015) <https://doi.org/10.1016/j.vetmic.2015.04.009> (although this is Salmonella), Smith et al. (2020) <https://doi.org/10.1111/1365-2664.13723>, Hald et al. (2016) <https://link.springer.com/article/10.1186/s13028-016-0192-9> (Campylobacter but suggests bird/livestock transmission), and Rivadeneira et al. (2016) <https://escholarship.org/uc/item/3733r6pf>. In contrast, the studies that have tried to examine the impacts of urbanization on foodborne pathogen prevalence in birds have largely failed to see an effect, e.g., % urban didn't matter in Smith et al. (2020) <https://doi.org/10.1111/1365-2664.13723>, Rouffaer et al. (2016) (<https://journals.plos.org/plosone/article?id=10.1371/journal.pone.0155366>), Brobey, Kucknoor & Armacost (2017) (<https://doi.org/10.1637/11607-020617-RegR>), nor Hamer, Lehrer & Magle (2012) (<https://doi.org/10.1111/j.1863-2378.2012.01462.x>) (but Hernandez et al. (2016) did see an impact of urbanization on white ibis; <https://doi.org/10.1371/journal.pone.0164402>).

I like the title.

Abstract:

L16-18: There are not too many cases where wildlife were definitively linked to outbreaks of E. coli. I would say, "...are sometimes implicated as the source of pathogenic..." or "have been implicated." The primary ones that I am aware of were the 2006 spinach outbreak in California in

which they found matching isolates between pigs and cattle near the field, but the final CDC report said water flooding could have been responsible. There was also an Oregon outbreak thought to be caused by deer. (These are mentioned at some point in the article). In any case, I think the language is a bit strong at this sentence.

Introduction:

There are no citations from lines 32-46 to support the statements made. The statements themselves seem accurate, but the authors should add citations.

L49-51: The authors never specified any detail on how studies were acquired. Was there a systematic search? What were search terms used? Did they search literature that came up in their search for more references? How can the authors be sure their protocol sufficiently acquired studies? For example, one citation missing in references that is very relevant is Navarro-Gonzalez et al. (2020) <https://aem.asm.org/content/86/3/e01678-19.abstract>

L51-53: Here, I will explain why I do not think the findings are novel enough for Proceedings B. I think the review is very valuable but would fit better at a discipline-specific journal. For example, Smith et al. (2020) <https://onlinelibrary.wiley.com/doi/full/10.1111/brv.12581> outlined points 1 and 3 (inconsistency between studies and bias and underrepresentation of wild bird hosts studied). The geographic bias outlined in the paper is an issue in the broader literature across topics of investigation, e.g., Clarke et al. (2017) <https://doi.org/10.1016/j.tree.2017.02.012> Greater emphasis on the AMR and genetic diversity aspects of the review, to me, would stress the novel aspects because I do not believe they are well-covered for *E. coli* in a review.

L72-88: I think the results emphasized at L72-88 could go in an appendix in favor of highlighting information on wildlife-livestock transmission and methods to acquire literature. The information presented has largely been covered elsewhere.

L81-88: This result has previously been identified by Smith et al. (2020) *Biol Rev* for birds. I am unsure if anyone has robustly summarized this for mammals.

Section starting at L113: I think this section is a more novel component to review. I would emphasize these results. I'd also suggest mentioning more of the role of antibiotic use in livestock in the carriage of AMR genes by birds.

L125: I would add a reference to the WHO report page

L143-144: How prevalent?

L176: Add references to support statements

L195: I am only somewhat familiar with sequence databases, but I am not entirely sure this is true. There is PubMLST <https://pubmlst.org/organisms> (seems more promising) and PulseNet <https://www.cdc.gov/pulsenet/index.html>

L199: Add reference to support that wildlife are implicated as sources of contamination

L204-206: Generally, livestock, worker sanitation, and water are considered bigger risk factors for food safety, with wildlife considered a pretty minor contributor. E.g., Parker et al. (2012)

<https://link.springer.com/article/10.1007/s10460-012-9360-3> or Park et al. (2013)

<https://aem.asm.org/content/79/14/4347.short>

L207: "birds and insectivores" -> "insectivorous birds and bats" or something similar. Also, add reference to support benefit of ecosystem services.

L217-218: I would say this is an unfair assessment. There are protocols in place, but it is challenging.

L221: Reference? The Park et al. article mentioned above would work here.

L246-249: I suggest breaking this into two sentences. I believe "effecting" should be "affecting" or "impacting"

Conclusion:

L260-264: I think that comparing COVID spread to *E. coli* spread is unrealistic because *E. coli* is a fecal-oral pathogen whereas COVID is respiratory. According to Batz et al. (2012),

<https://doi.org/10.4315/0362-028X.JFP-11-418>, there are ~65,153 cases of O157:H7 and 112,752 cases of non-O157 in the US each year. According to the WHO, there are 8,403,121 confirmed cases of COVID in the US at this time (a partial year). Therefore, I think due to the very different transmission pathways, *E. coli* is unlikely to be comparable to COVID, even with AMR genes.

L279: I believe "effect" here should also be "affect"

Supporting information:

I would prefer one merged file instead of having to download 4

Review form: Reviewer 2

Recommendation

Accept with minor revision (please list in comments)

Scientific importance: Is the manuscript an original and important contribution to its field?

Excellent

General interest: Is the paper of sufficient general interest?

Good

Quality of the paper: Is the overall quality of the paper suitable?

Good

Is the length of the paper justified?

Yes

Should the paper be seen by a specialist statistical reviewer?

No

Do you have any concerns about statistical analyses in this paper? If so, please specify them explicitly in your report.

No

It is a condition of publication that authors make their supporting data, code and materials available - either as supplementary material or hosted in an external repository. Please rate, if applicable, the supporting data on the following criteria.

Is it accessible?

N/A

Is it clear?

N/A

Is it adequate?

N/A

Do you have any ethical concerns with this paper?

No

Comments to the Author

Review RSPB-2020-2526

General comments

This manuscript is a review of the (rare; less than 100 articles) existing literature on E. coli studies in wildlife populations in situ. It provides information about the geographic and host distribution, the AMR found and its pathogenicity. The authors rightfully highlight that there is little information available on this topic, that there is a risk for global public health and that the research community should focus on this compartment. Finally, the authors conclude by

presenting some research priorities.

The manuscript is timely, of interest and well-written. I have a few comments that could improve its quality.

Firstly, I was expecting to see a summary of the main phylogroups found in wild hosts compared to what is found in other domestic and human hosts. There is no comparative approach for non-specialists. For example, I would have liked to know if wildlife hosts have a “phylogroup profile” closer to livestock (predominance of B1)? Or to human? Or if this phylogroup profile was more impacted by geography? By animal order (mammals vs. birds)? By latitude as biodiversity? I guess one cannot answer those questions with the current data, but these would be interesting questions to raise in the manuscript.

Secondly, regarding AMR *E. coli* in wildlife: there is not a clear section presenting AMR as being the product of a natural process and (pre)existing in nature. Today we have the tools to differentiate between natural and anthropogenic AMR strains. What we observe in the few existing studies is what you describe as AMR being mainly anthropogenic and percolating into the wild. Why did you not take that perspective? (you still mention their “prevalence in natural environments” L136). In addition, the evolution of AMR genes in the wild merits some comments: a priori they should be counter-selected in environments with low AB pressure but some studies indicated that there could be AMR genes with no cost to its host.

Thirdly, I would like the authors to question the relative risk of AMR in wildlife hosts on public health. Of course, there is a risk. But in a resource-limited research context, is this research a priority? I would expect the authors to discuss the issue in the broader AMR context and global public health. For example, from a global public health perspective, who are the humans most impacted by AMR principally (us farmers and consumers vs hospital patients)? Where are they living? Do we know if they are greatly exposed to AMR from wildlife? This question is important regarding the global fight against AMR. Do we really need to invest into AMR in wildlife research or we should prioritize other AMR research sectors?

Then, your article is written mainly with the perspective of the impact on global public health. Dealing with wild hosts, you could also in the introduction and maybe in the conclusion discuss potential biodiversity conservation impacts at host level but also at genetic diversity level? Same remark for the impact of AMR from and on livestock production?

Finally, I was expecting a few lines on the difficulty and requirements linked to sampling *E. coli* in wildlife hosts, especially in understudied countries where sampling, storing and sending (practically and legally) samples from wildlife could be difficult. This can also explain the paucity and the lack of quality of data available and the need to collect relevant meta-data when working with wildlife (e.g. species, size of group, estimating age, estimation of the wildlife/livestock/human interface and interactions with other wild species etc.).

Specific comments

- There is no information about how the literature review was implemented. Could you provide briefly the process? For example, which keywords did you choose to find the <100 wild animal studies out of the > 2 million *E. coli* studies?

- In addition you mention in the abstract the “mammalian gut microbiome”, in the title “wild animal”, then you talk about mammals, birds and in Figure 4 there is also a fly. Did you limit your review to some orders? (no mention of reptiles because there is no study on reptiles or because you did not review them?). The reader would need clarification on this.

- L24-L25: Abstract: “because of the strong implications for global public health”: are you confident that the role of “wildlife” is relatively that important? Do we have proofs for that? (I

mean a significant amount of proof or we don't know the risk but it could be important?). That's the main argument that was said to me when I tried to justify for funding for AMR in wildlife (and did not get it!).

- L37-38: "identifying sources of agricultural contamination of pathogenic E. coli": specify "natural" or "wild" E. coli because at first reading I thought you were talking about contamination from agricultural origin into the wild.

- L46: you mention "invertebrates" but you don't give any reference.

- L103-105: "The first study to examine E. coli in wild animals...": I don't understand the sentence. There is a word missing or clarify. What level of diversity are you talking about? Phylogroup I guess?

- L138-140: and this leads to your concept of "melting pots". I agree with this concept but according to your presentation the global process is: AMR is produced in anthropological systems (animal production, human populations, humanly-impacted environment), then this AMR is hosted in wildlife where it can be maintained, evolve and spill back to human or production animals. In your melting pot, "natural" AMR does not seem to play a role (and it is reinforced by my point above). Don't you think that the mixing of anthropogenic and natural AMR is a potential threat for "novelty"? If yes, maybe you should have it clearly said in this section. Do we have any indication of a pathogenic strain that have evolved in wildlife, spilled back into humans or livestock with a higher pathogenicity? If no, and if I agree that this is a risk, you should relativize this risk.

- L278-286: you suggest rightly that we can "easily" follow the spread of AMR into wildlife from a human origin (because there should not be a lot of AMR in the wild); and then you say that there is already so much AMR in the wild that the risk of spill back to humans and livestock is important and we have difficulty to trace it. There is a small paradox here that you need to discuss. In this paragraph on routes of transmission, you don't mention the wildlife/livestock/interfaces that are the places where spillover and spill-back of E.coli strain towards and from the wild occur. Maybe that research should also focus on some of these interfaces to better understand the ecology of E. coli transmission?

- L143: ESBL-E.coli: maybe explain a bit more what this strain is and that it has a human origin for the RSPB non-specialized readership.

- L210: "(...) produce production (...)": awkward.

- L248: "effecting"?

- L260-261: the reference to COVID-19 is maybe a bit opportunistic and I don't think it brings a lot to the conclusion, except if you take the time to explain the challenges at the nexus agriculture/biodiversity/health and the global changes that impact our relationship with the nature, which COVID-19 is a symptom. (in the last 6 months, 100% of the articles I review mention the COVID-19 crisis and I don't think it is necessary). Maybe a reference to some integrated approaches to health would be more suitable?

Figure 2(a): the background "grey" does not differ much from the "light grey" countries making it difficult to locate countries with only 1 study.

Decision letter (RSPB-2020-2526.R0)

11-Nov-2020

Dear Dr Lagerstrom:

I am writing to inform you that your manuscript RSPB-2020-2526 entitled "The under-investigated wild side of *Escherichia coli*: genetic diversity, pathogenicity, and antimicrobial resistance in wild animal hosts" has, in its current form, been rejected for publication in Proceedings B.

The referees diverge in their opinions, but they both agree that it is well written, timely and on an important topic. That's certainly my view too. However, referee 1 feels that there just aren't enough novel insights for a Proceedings B review, while referee 2 feels that, with suitable revision, it would meet the grade. So, it's one of those situations that authors (and editors) hate - a subjective judgement by the editor about novelty and impact. Both referees do have substantive criticisms, but I can see that you might be able to address these. Certainly referee 1's point about the lack of information on how the literature was searched is crucial -- unless the search was systematic, using replicable criteria, it is impossible to judge how robust the patterns described are. However, again, this could be fixed. The issue whether there are enough novel ideas is a concern and not one I can easily judge - I need to take the referees' word for this - but I do see an overlap with Smith et al. (2020) which, given that journal, could go into more depth than a Proceedings B review. That said, there is a place for shorter focused reviews (if there wasn't, we wouldn't do them!) and so, with this in mind, I would be willing to consider a resubmission, provided the comments of the referees are fully addressed. However please note that this is not a provisional acceptance.

- 1) A 'response to referees' document including details of how you have responded to the comments, and the adjustments you have made.
- 2) A clean copy of the manuscript and one with 'tracked changes' indicating your 'response to referees' comments document.
- 3) Line numbers in your main document.
- 4) Please read our data sharing policies to ensure that you meet our requirements <https://royalsociety.org/journals/authors/author-guidelines/#data>.

Best wishes,
Innes Cuthill

Prof. Innes Cuthill
Reviews Editor, Proceedings B

Reviewer(s)' Comments to Author:

Referee: 1

Comments to the Author(s)

Manuscript RSPB-2020-2526 is an interesting and well-written review highlighting the sparse literature available to date on *E. coli* in wildlife, with an emphasis on antimicrobial resistance. I have mostly minor comments throughout. My primary concern with publication of the manuscript in Proceedings B is that many of the components are highlighted elsewhere in the literature. I believe it will make a nice contribution to the literature but isn't a major enough advancement for Proceedings B.

My primary big comment for the authors in revising the manuscript is that I think the review should have greater emphasis on livestock-wildlife transmission. For example, Fig. 3 focuses on an axis of increasing human population density and urban impact. However, exchange of *E. coli* (and other foodborne pathogens) often occurs at the wildlife-livestock interface rather than urban interface and can also be the source of AMR genes. Michele Jay-Russell's lab has quite a lot of literature on the matter, but to give some specific examples of a few refs of many that show or suggest wildlife-livestock transmission of foodborne pathogens: Carlson et al. (2015) <https://doi.org/10.1016/j.vetmic.2015.04.009> (although this is *Salmonella*), Smith et al. (2020) <https://doi.org/10.1111/1365-2664.13723>, Hald et al. (2016) <https://link.springer.com/article/10.1186/s13028-016-0192-9> (*Campylobacter* but suggests bird/livestock transmission), and Rivadeneira et al. (2016) <https://escholarship.org/uc/item/3733r6pf>. In contrast, the studies that have tried to examine the impacts of urbanization on foodborne pathogen prevalence in birds have largely failed to see an effect, e.g., % urban didn't matter in Smith et al. (2020) <https://doi.org/10.1111/1365-2664.13723>, Rouffaer et al. (2016) (<https://journals.plos.org/plosone/article?id=10.1371/journal.pone.0155366>), Brobey, Kucknoor & Armacost (2017) (<https://doi.org/10.1637/11607-020617-RegR>), nor Hamer, Lehrer & Magle (2012) (<https://doi.org/10.1111/j.1863-2378.2012.01462.x>) (but Hernandez et al. (2016) did see an impact of urbanization on white ibis; <https://doi.org/10.1371/journal.pone.0164402>).

I like the title.

Abstract:

L16-18: There are not too many cases where wildlife were definitively linked to outbreaks of *E. coli*. I would say, "...are sometimes implicated as the source of pathogenic..." or "have been implicated." The primary ones that I am aware of were the 2006 spinach outbreak in California in which they found matching isolates between pigs and cattle near the field, but the final CDC report said water flooding could have been responsible. There was also an Oregon outbreak thought to be caused by deer. (These are mentioned at some point in the article). In any case, I think the language is a bit strong at this sentence.

Introduction:

There are no citations from lines 32-46 to support the statements made. The statements themselves seem accurate, but the authors should add citations.

L49-51: The authors never specified any detail on how studies were acquired. Was there a systematic search? What were search terms used? Did they search literature that came up in their search for more references? How can the authors be sure their protocol sufficiently acquired studies? For example, one citation missing in references that is very relevant is Navarro-Gonzalez et al. (2020) <https://aem.asm.org/content/86/3/e01678-19.abstract>

L51-53: Here, I will explain why I do not think the findings are novel enough for Proceedings B. I think the review is very valuable but would fit better at a discipline-specific journal. For example, Smith et al. (2020) <https://onlinelibrary.wiley.com/doi/full/10.1111/brv.12581> outlined points 1 and 3 (inconsistency between studies and bias and underrepresentation of wild bird hosts

studied). The geographic bias outlined in the paper is an issue in the broader literature across topics of investigation, e.g., Clarke et al. (2017) <https://doi.org/10.1016/j.tree.2017.02.012>
Greater emphasis on the AMR and genetic diversity aspects of the review, to me, would stress the novel aspects because I do not believe they are well-covered for E. coli in a review.

L72-88: I think the results emphasized at L72-88 could go in an appendix in favor of highlighting information on wildlife-livestock transmission and methods to acquire literature. The information presented has largely been covered elsewhere.

L81-88: This result has previously been identified by Smith et al. (2020) *Biol Rev* for birds. I am unsure if anyone has robustly summarized this for mammals.

Section starting at L113: I think this section is a more novel component to review. I would emphasize these results. I'd also suggest mentioning more of the role of antibiotic use in livestock in the carriage of AMR genes by birds.

L125: I would add a reference to the WHO report page

L143-144: How prevalent?

L176: Add references to support statements

L195: I am only somewhat familiar with sequence databases, but I am not entirely sure this is true. There is PubMLST <https://pubmlst.org/organisms> (seems more promising) and PulseNet <https://www.cdc.gov/pulsenet/index.html>

L199: Add reference to support that wildlife are implicated as sources of contamination

L204-206: Generally, livestock, worker sanitation, and water are considered bigger risk factors for food safety, with wildlife considered a pretty minor contributor. E.g., Parker et al. (2012)

<https://link.springer.com/article/10.1007/s10460-012-9360-3> or Park et al. (2013)

<https://aem.asm.org/content/79/14/4347.short>

L207: "birds and insectivores" -> "insectivorous birds and bats" or something similar. Also, add reference to support benefit of ecosystem services.

L217-218: I would say this is an unfair assessment. There are protocols in place, but it is challenging.

L221: Reference? The Park et al. article mentioned above would work here.

L246-249: I suggest breaking this into two sentences. I believe "effecting" should be "affecting" or "impacting"

Conclusion:

L260-264: I think that comparing COVID spread to E. coli spread is unrealistic because E. coli is a fecal-oral pathogen whereas COVID is respiratory. According to Batz et al. (2012), <https://doi.org/10.4315/0362-028X.JFP-11-418>, there are ~65,153 cases of O157:H7 and 112,752 cases of non-O157 in the US each year. According to the WHO, there are 8,403,121 confirmed cases of COVID in the US at this time (a partial year). Therefore, I think due to the very different transmission pathways, E. coli is unlikely to be comparable to COVID, even with AMR genes.

L279: I believe "effect" here should also be "affect"

Supporting information:

I would prefer one merged file instead of having to download 4

Referee: 2

Comments to the Author(s)

Review RSPB-2020-2526

General comments

This manuscript is a review of the (rare; less than 100 articles) existing literature on E. coli studies in wildlife populations in situ. It provides information about the geographic and host distribution, the AMR found and its pathogenicity. The authors rightfully highlight that there is little information available on this topic, that there is a risk for global public health and that the research community should focus on this compartment. Finally, the authors conclude by presenting some research priorities.

The manuscript is timely, of interest and well-written. I have a few comments that could improve its quality.

Firstly, I was expecting to see a summary of the main phylogroups found in wild hosts compared to what is found in other domestic and human hosts. There is no comparative approach for non-specialists. For example, I would have liked to know if wildlife hosts have a “phylogroup profile” closer to livestock (predominance of B1)? Or to human? Or if this phylogroup profile was more impacted by geography? By animal order (mammals vs. birds)? By latitude as biodiversity? I guess one cannot answer those questions with the current data, but these would be interesting questions to raise in the manuscript.

Secondly, regarding AMR *E. coli* in wildlife: there is not a clear section presenting AMR as being the product of a natural process and (pre)existing in nature. Today we have the tools to differentiate between natural and anthropogenic AMR strains. What we observe in the few existing studies is what you describe as AMR being mainly anthropogenic and percolating into the wild. Why did you not take that perspective? (you still mention their “prevalence in natural environments” L136). In addition, the evolution of AMR genes in the wild merits some comments: a priori they should be counter-selected in environments with low AB pressure but some studies indicated that there could be AMR genes with no cost to its host.

Thirdly, I would like the authors to question the relative risk of AMR in wildlife hosts on public health. Of course, there is a risk. But in a resource-limited research context, is this research a priority? I would expect the authors to discuss the issue in the broader AMR context and global public health. For example, from a global public health perspective, who are the humans most impacted by AMR principally (us farmers and consumers vs hospital patients)? Where are they living? Do we know if they are greatly exposed to AMR from wildlife? This question is important regarding the global fight against AMR. Do we really need to invest into AMR in wildlife research or we should prioritize other AMR research sectors?

Then, your article is written mainly with the perspective of the impact on global public health. Dealing with wild hosts, you could also in the introduction and maybe in the conclusion discuss potential biodiversity conservation impacts at host level but also at genetic diversity level? Same remark for the impact of AMR from and on livestock production?

Finally, I was expecting a few lines on the difficulty and requirements linked to sampling *E. coli* in wildlife hosts, especially in understudied countries where sampling, storing and sending (practically and legally) samples from wildlife could be difficult. This can also explain the paucity and the lack of quality of data available and the need to collect relevant meta-data when working with wildlife (e.g. species, size of group, estimating age, estimation of the wildlife/livestock/human interface and interactions with other wild species etc.).

Specific comments

- There is no information about how the literature review was implemented. Could you provide briefly the process? For example, which keywords did you choose to find the <100 wild animal studies out of the > 2 million *E. coli* studies?

- In addition you mention in the abstract the “mammalian gut microbiome”, in the title “wild animal”, then you talk about mammals, birds and in Figure 4 there is also a fly. Did you limit your review to some orders? (no mention of reptiles because there is no study on reptiles or because you did not review them?). The reader would need clarification on this.

- L24-L25: Abstract: “because of the strong implications for global public health”: are you confident that the role of “wildlife” is relatively that important? Do we have proofs for that? (I mean a significant amount of proof or we don't know the risk but it could be important?). That's

the main argument that was said to me when I tried to justify for funding for AMR in wildlife (and did not get it!).

- L37-38: “identifying sources of agricultural contamination of pathogenic E. coli”: specify “natural” or “wild” E. coli because at first reading I thought you were talking about contamination from agricultural origin into the wild.

- L46: you mention “invertebrates” but you don’t give any reference.

- L103-105: “The first study to examine E. coli in wild animals...”: I don’t understand the sentence. There is a word missing or clarify. What level of diversity are you talking about? Phylogroup I guess?

- L138-140: and this leads to your concept of “melting pots”. I agree with this concept but according to your presentation the global process is: AMR is produced in anthropological systems (animal production, human populations, humanly-impacted environment), then this AMR is hosted in wildlife where it can be maintained, evolve and spill back to human or production animals. In your melting pot, “natural” AMR does not seem to play a role (and it is reinforced by my point above). Don’t you think that the mixing of anthropogenic and natural AMR is a potential threat for “novelty”? If yes, maybe you should have it clearly said in this section. Do we have any indication of a pathogenic strain that have evolved in wildlife, spilled back into humans or livestock with a higher pathogenicity? If no, and if I agree that this is a risk, you should relativize this risk.

- L278-286: you suggest rightly that we can “easily” follow the spread of AMR into wildlife from a human origin (because there should not be a lot of AMR in the wild); and then you say that there is already so much AMR in the wild that the risk of spill back to humans and livestock is important and we have difficulty to trace it. There is a small paradox here that you need to discuss. In this paragraph on routes of transmission, you don’t mention the wildlife/livestock/interfaces that are the places where spillover and spill-back of E.coli strain towards and from the wild occur. Maybe that research should also focus on some of these interfaces to better understand the ecology of E. coli transmission?

- L143: ESBL-E.coli: maybe explain a bit more what this strain is and that it has a human origin for the RSPB non-specialized readership.

- L210: “(...) produce production (...)”: awkward.

- L248: “effecting”?

- L260-261: the reference to COVID-19 is maybe a bit opportunistic and I don’t think it brings a lot to the conclusion, except if you take the time to explain the challenges at the nexus agriculture/biodiversity/health and the global changes that impact our relationship with the nature, which COVID-19 is a symptom. (in the last 6 months, 100% of the articles I review mention the COVID-19 crisis and I don’t think it is necessary). Maybe a reference to some integrated approaches to health would be more suitable?

Figure 2(a): the background “grey” does not differ much from the “light grey” countries making it difficult to locate countries with only 1 study.

Author's Response to Decision Letter for (RSPB-2020-2526.R0)

See Appendix A.

RSPB-2021-0399.R0

Review form: Reviewer 1

Recommendation

Accept with minor revision (please list in comments)

Scientific importance: Is the manuscript an original and important contribution to its field?

Good

General interest: Is the paper of sufficient general interest?

Acceptable

Quality of the paper: Is the overall quality of the paper suitable?

Excellent

Is the length of the paper justified?

Yes

Should the paper be seen by a specialist statistical reviewer?

No

Do you have any concerns about statistical analyses in this paper? If so, please specify them explicitly in your report.

No

It is a condition of publication that authors make their supporting data, code and materials available - either as supplementary material or hosted in an external repository. Please rate, if applicable, the supporting data on the following criteria.

Is it accessible?

Yes

Is it clear?

Yes

Is it adequate?

No

Do you have any ethical concerns with this paper?

No

Comments to the Author

The authors did an overall good job incorporating the suggestions from the first revision. Below, I made some additional, mostly minor, suggestions. I think the article will be of interest to people studying E. coli and wildlife-associated food safety issues.

Generally, the paper is well-written. I suggest trying to reduce some of the really long sentences in the new sections. Many have > 50 words and are hard to follow.

Abstract:

L13-14: I don't think this sentence does a lot to set up the paper, and I think it could be removed. The introductory material in the abstract takes up quite a bit of space that could be traded off to highlight findings from the review. I think it'd help the article's eventual impact since,

unfortunately, a lot of people don't read past abstracts.

L27-28: Given reviewer 2's comments, it seems more appropriate to say, "...especially in light of the potentially strong implications..."

Main text:

The additional text in the introduction sets up the paper better.

L69-70: Citations to anchor the statements (that E. coli causes a large public health burden and ES provide high economic value) would be good.

L85: "In order to" -> "to"

L117-128: If leaving this in as a major finding, I suggest adding in some of your rationale in the response to reviewers that better explains the importance of the information: "Geographic biases in where E. coli has been studied in wild animal populations poses issues unique to the topic because it disallows answering questions important to protecting global public health such as; is the distribution of E. coli geographically-determined, host specific or ubiquitous? Are there higher rates of AMR in countries with weaker controls on antibiotic use in both medical and agricultural sectors? Are rates of clinical cases of AMR pathogenic E. coli higher in countries where people have more contact with wild animals?"

Overall, I think the current paragraph doesn't set up the rationale/importance as well as how you explained it in the response.

L154: For more general audiences, specifying what "H" is would be helpful

L182-185: Is there a published/peer-reviewed example you could use here instead? An unpublished report isn't the best evidence.

L194: Should this say, "antibiotics used in medicine"? I had to re-read this a few times.

L195: "Piece" -> "pieces"

L194-221: The sentences in this new section are pretty long. I suggest trying to break them up to be ~25 words or less each to make it easier to parse.

L221: I suggest being less definitive with "the risk of such an occurrence is real" since no one has documented it. I think your reply to reviewers was great as to why that is very hard to do. However, it's pretty strong language in text. Perhaps, "the risk of such an occurrence is nonzero" or "such an occurrence could theoretically happen." You could also rephrase the sentence to say something like, "We have not yet documented a pathogenic strain that evolved in such a way in wildlife and spilled back into humans or livestock with a higher pathogenicity. However, such approaches are highly impractical, and it would be nearly impossible to document in the wild."

L297-300: I think this result is a little more complex than stated. The semi-natural land cover clearing can have impacts on the wildlife species that are found on the farms. These species can vary in reservoir competence. Clearing of semi-natural land cover, may, therefore shift the community towards better hosts (e.g., Smith et al. 2020 JAE). I might suggest a change like, "Despite these environmentally-harmful measures, prevalence of pathogenic E. coli actually increased in many cases by more than an order of magnitude between 2007 and 2013, indicating that removal of semi-natural land cover may be ineffective (Karp et al. 2015 PNAS; Smith et al. 2020 JAE)." Then, you could emphasize at L300-303 that these practices can favor species that are better hosts (e.g., European starlings). I'd suggest a modification like, "Exclusion or elimination of native species on and surrounding farmland, such as bees, insectivorous birds and bats, via

habitat destruction, trapping or killing, could potentially lead to incalculable losses of ecosystem services these wild species provide. Destruction of native land cover may also shift biological communities towards dominance by competent hosts, counterintuitively exacerbating food safety risks (Smith et al. 2020 JAE).” Then, in the last sentence, you could mention that it’s also important to understand the contribution of individual species, which has an implication for management and risk. I’d expect birds with greater degrees of synanthropy to have greater exposure/problems, for example, so farming environments that favor these species are likely riskier for public health.

<https://besjournals.onlinelibrary.wiley.com/doi/full/10.1111/1365-2664.13723>
<https://doi.org/10.1073/pnas.1508435112>

L313-314: I might say, “an important gap in monitoring of foodborne pathogen transmission” or “an important gap in monitoring of enteric pathogen transmission” since they thought the cases were due to petting animals rather than consuming food.

L319: “In order to” -> “to”

L321-322: There are a few references that discuss these issues, and adding a citation to anchor the statement would be good. Already cited in the manuscript was the Beretti and Stuart example, but Danny Karp and colleagues have also published a bit on this tension. A recent and in depth one: <https://link.springer.com/article/10.1007/s10460-020-10123-8>

L344: The paragraph as edited might be more accurately subtitled, “Carried by birds” or “Movement by birds”

L355: I assume you’ll need the scientific name for specific species mentioned like Glaucous-winged gull

L361: “It was found that” reads a little awkwardly. Perhaps revise to something like, “Rivadeneira et al. (74) found that wild birds can carry STEC between CAFOs (concentrated animal feeding operations) and leafy green fields; about 5% of birds in their study carried STEC (74)”

At 365, rather than “they identified,” I suggest, “LeJeune et al. (75) identified”

Conclusion

L:415-421: I think it is of note to also mention it’s important to sample a wide-range of species to tie back into conservation and ecosystem services that is brought up throughout. Are species of conservation concern frequently harboring/spreading virulence and AMR genes or do they rarely harbor/spread them? If species that are key transmitters differ from those that are of conservation concern and those that provide services, then that suggests managing for disease, conservation, and ecosystem services aren’t at odds.

L432: I’d spell out Democratic Republic of the Congo as not all readers may pick up on what the DRC is

L432: “in order to” -> “to”

Figures:

Fig. 2: “Countries in grey were not represented in this literature review” makes it sound like you didn’t include them on purpose. Perhaps say, “No studies fitting our inclusion criteria were found in countries in grey.” Also, the numbers in Panel B are hard to read/see. It might work better if you put the numbers in the white space with a line going to the country.

Table S1. "References follow the table" would be clearer.

Table S3. It would be good to check over the nomenclature and consistency in what words are capitalized, etc. on Table S3. It would be helpful to indicate what you followed for your common and scientific names and the date you searched for taxa since things change rapidly and differ between, e.g., IUCN and American Ornithological Society. For example, you use "common starling" over "European starling." I think whatever you used is fine, but it would be good to know who you are following. I suggest italicizing the scientific names (assuming this is journal policy).

Also, it is an interesting table but would be more informative/impactful if you had a label for if it is a bird or mammal. It would also be more useful to readers if you added the references that include the species to make it easy to locate individual ones, if say, you wanted to learn more about rock pigeons.

Lastly, upon reading the table, I noticed some specific typos or issues, but I suggest carefully going back over it for others.

Specific things I found: I'd make "magpie" "Eurasian magpie" since there are many magpie species.

Calling *Aegypius monachus* "black vulture" over Cinereous vulture is also confusing since *Coragyps atratus* is the "black vulture," according to the American Ornithological Society official common names (and American black vulture in IUCN). *Coragyps atratus* is also listed separately in the table. I think the original reference used "black vulture," but I'd try and standardize to IUCN or something similar.

House sparrow should be "*Passer domesticus*."

American redstart is missing the "t" on "redstart."

"Cotte duck" is missing a scientific name.

"Hoepoe" is a "hoopoe." This appears to be Eurasian hoopoe specifically.

Haliaeetus albicilla is labeled "sea eagles," which seems to be referencing a genus *Haliaeetus*, but then white-tailed eagle appears to be the specific eagle mentioned.

"Pygmy" is misspelled in "western pygmy possum."

Decision letter (RSPB-2021-0399.R0)

08-Mar-2021

Dear Dr Lagerstrom

I am pleased to inform you that your manuscript RSPB-2021-0399 entitled "The under-investigated wild side of *Escherichia coli*: genetic diversity, pathogenicity, and antimicrobial resistance in wild animal hosts" has been accepted for publication in *Proceedings B*.

The referee thinks you have done a good job of revision and has recommended publication, but also suggest some further minor revisions to your manuscript. The list isn't short, but all the changes are straightforward. Therefore, I invite you to respond to the referee's comments and revise your manuscript. Because the schedule for publication is very tight, it is a condition of

publication that you submit the revised version of your manuscript within 7 days. If you do not think you will be able to meet this date please let us know.

If you wish to submit your data to Dryad (<http://datadryad.org/>) and have not already done so you can submit your data via this link [http://datadryad.org/submit?journalID=RSPB&manu=\(Document not available\)](http://datadryad.org/submit?journalID=RSPB&manu=(Document+not+available)) which will take you to your unique entry in the Dryad repository. If you have already submitted your data to dryad you can make any necessary revisions to your dataset by following the above link. Please see <https://royalsociety.org/journals/ethics-policies/data-sharing-mining/> for more details.

Best wishes,
Innes Cuthill

Prof. Innes Cuthill
Reviews Editor, Proceedings B
mailto: proceedingsb@royalsociety.org

Reviewer(s)' Comments to Author:
Referee: 1

Comments to the Author(s)

The authors did an overall good job incorporating the suggestions from the first revision. Below, I made some additional, mostly minor, suggestions. I think the article will be of interest to people studying *E. coli* and wildlife-associated food safety issues.

Generally, the paper is well-written. I suggest trying to reduce some of the really long sentences in the new sections. Many have > 50 words and are hard to follow.

Abstract:

L13-14: I don't think this sentence does a lot to set up the paper, and I think it could be removed. The introductory material in the abstract takes up quite a bit of space that could be traded off to highlight findings from the review. I think it'd help the article's eventual impact since, unfortunately, a lot of people don't read past abstracts.

L27-28: Given reviewer 2's comments, it seems more appropriate to say, "...especially in light of the potentially strong implications..."

Main text:

The additional text in the introduction sets up the paper better.

L69-70: Citations to anchor the statements (that *E. coli* causes a large public health burden and ES provide high economic value) would be good.

L85: "In order to" -> "to"

L117-128: If leaving this in as a major finding, I suggest adding in some of your rationale in the response to reviewers that better explains the importance of the information: “Geographic biases in where *E. coli* has been studied in wild animal populations poses issues unique to the topic because it disallows answering questions important to protecting global public health such as; is the distribution of *E. coli* geographically-determined, host specific or ubiquitous? Are there higher rates of AMR in countries with weaker controls on antibiotic use in both medical and agricultural sectors? Are rates of clinical cases of AMR pathogenic *E. coli* higher in countries where people have more contact with wild animals?”

Overall, I think the current paragraph doesn’t set up the rationale/importance as well as how you explained it in the response.

L154: For more general audiences, specifying what “H” is would be helpful

L182-185: Is there a published/peer-reviewed example you could use here instead? An unpublished report isn’t the best evidence.

L194: Should this say, “antibiotics used in medicine”? I had to re-read this a few times.

L195: “Piece” -> “pieces”

L194-221: The sentences in this new section are pretty long. I suggest trying to break them up to be ~25 words or less each to make it easier to parse.

L221: I suggest being less definitive with “the risk of such an occurrence is real” since no one has documented it. I think your reply to reviewers was great as to why that is very hard to do. However, it’s pretty strong language in text. Perhaps, “the risk of such an occurrence is nonzero” or “such an occurrence could theoretically happen.” You could also rephrase the sentence to say something like, “We have not yet documented a pathogenic strain that evolved in such a way in wildlife and spilled back into humans or livestock with a higher pathogenicity. However, such approaches are highly impractical, and it would be nearly impossible to document in the wild.”

L297-300: I think this result is a little more complex than stated. The semi-natural land cover clearing can have impacts on the wildlife species that are found on the farms. These species can vary in reservoir competence. Clearing of semi-natural land cover, may, therefore shift the community towards better hosts (e.g., Smith et al. 2020 JAE). I might suggest a change like, “Despite these environmentally-harmful measures, prevalence of pathogenic *E. coli* actually increased in many cases by more than an order of magnitude between 2007 and 2013, indicating that removal of semi-natural land cover may be ineffective (Karp et al. 2015 PNAS; Smith et al. 2020 JAE).” Then, you could emphasize at L300-303 that these practices can favor species that are better hosts (e.g., European starlings). I’d suggest a modification like, “Exclusion or elimination of native species on and surrounding farmland, such as bees, insectivorous birds and bats, via habitat destruction, trapping or killing, could potentially lead to incalculable losses of ecosystem services these wild species provide. Destruction of native land cover may also shift biological communities towards dominance by competent hosts, counterintuitively exacerbating food safety risks (Smith et al. 2020 JAE).” Then, in the last sentence, you could mention that it’s also important to understand the contribution of individual species, which has an implication for management and risk. I’d expect birds with greater degrees of synanthropy to have greater exposure/problems, for example, so farming environments that favor these species are likely riskier for public health.

<https://besjournals.onlinelibrary.wiley.com/doi/full/10.1111/1365-2664.13723>
<https://doi.org/10.1073/pnas.1508435112>

L313-314: I might say, “an important gap in monitoring of foodborne pathogen transmission” or “an important gap in monitoring of enteric pathogen transmission” since they thought the cases were due to petting animals rather than consuming food.

L319: “In order to” -> “to”

L321-322: There are a few references that discuss these issues, and adding a citation to anchor the statement would be good. Already cited in the manuscript was the Beretti and Stuart example, but Danny Karp and colleagues have also published a bit on this tension. A recent and in depth one: <https://link.springer.com/article/10.1007/s10460-020-10123-8>

L344: The paragraph as edited might be more accurately subtitled, “Carried by birds” or “Movement by birds”

L355: I assume you’ll need the scientific name for specific species mentioned like Glaucous-winged gull

L361: “It was found that” reads a little awkwardly. Perhaps revise to something like, “Rivadeneira et al. (74) found that wild birds can carry STEC between CAFOs (concentrated animal feeding operations) and leafy green fields; about 5% of birds in their study carried STEC (74)”

At 365, rather than “they identified,” I suggest, “LeJeune et al. (75) identified”

Conclusion

L:415-421: I think it is of note to also mention it’s important to sample a wide-range of species to tie back into conservation and ecosystem services that is brought up throughout. Are species of conservation concern frequently harboring/spreading virulence and AMR genes or do they rarely harbor/spread them? If species that are key transmitters differ from those that are of conservation concern and those that provide services, then that suggests managing for disease, conservation, and ecosystem services aren’t at odds.

L432: I’d spell out Democratic Republic of the Congo as not all readers may pick up on what the DRC is

L432: “in order to” -> “to”

Figures:

Fig. 2: “Countries in grey were not represented in this literature review” makes it sound like you didn’t include them on purpose. Perhaps say, “No studies fitting our inclusion criteria were found in countries in grey.” Also, the numbers in Panel B are hard to read/see. It might work better if you put the numbers in the white space with a line going to the country.

Table S1. “References follow the table” would be clearer.

Table S3. It would be good to check over the nomenclature and consistency in what words are capitalized, etc. on Table S3. It would be helpful to indicate what you followed for your common and scientific names and the date you searched for taxa since things change rapidly and differ between, e.g., IUCN and American Ornithological Society. For example, you use “common starling” over “European starling.” I think whatever you used is fine, but it would be good to know who you are following. I suggest italicizing the scientific names (assuming this is journal policy).

Also, it is an interesting table but would be more informative/impactful if you had a label for if it is a bird or mammal. It would also be more useful to readers if you added the references that

include the species to make it easy to locate individual ones, if say, you wanted to learn more about rock pigeons.

Lastly, upon reading the table, I noticed some specific typos or issues, but I suggest carefully going back over it for others.

Specific things I found: I'd make "magpie" "Eurasian magpie" since there are many magpie species.

Calling *Aegypius monachus* "black vulture" over Cinereous vulture is also confusing since *Coragyps atratus* is the "black vulture," according to the American Ornithological Society official common names (and American black vulture in IUCN). *Coragyps atratus* is also listed separately in the table. I think the original reference used "black vulture," but I'd try and standardize to IUCN or something similar.

House sparrow should be "*Passer domesticus*."

American redstart is missing the "t" on "redstart."

"Cotte duck" is missing a scientific name.

"Hoepoe" is a "hoopoe." This appears to be Eurasian hoopoe specifically.

Haliaeetus albicilla is labeled "sea eagles," which seems to be referencing a genus *Haliaeetus*, but then white-tailed eagle appears to be the specific eagle mentioned.

"Pygmy" is misspelled in "western pygmy possum."

Author's Response to Decision Letter for (RSPB-2021-0399.R0)

See Appendix B.

Decision letter (RSPB-2021-0399.R1)

18-Mar-2021

Dear Ms Lagerstrom

I am pleased to inform you that your manuscript entitled "The under-investigated wild side of *Escherichia coli*: genetic diversity, pathogenicity, and antimicrobial resistance in wild animals" has been accepted for publication in Proceedings B.

If you are likely to be away from e-mail contact during this period, let us know. Due to rapid publication and an extremely tight schedule, if comments are not received, we may publish the paper as it stands.

Data Accessibility section

Open access

You are invited to opt for open access via our author pays publishing model. Payment of open access fees will enable your article to be made freely available via the Royal Society website as soon as it is ready for publication. For more information about open access publishing please visit our website at http://royalsocietypublishing.org/site/authors/open_access.xhtml.

The open access fee is £1,700 per article (plus VAT for authors within the EU). If you wish to opt for open access then please let us know as soon as possible.

Paper charges

Sincerely,

Proceedings B

Appendix A

Feb 16, 2021

Response to referees

Manuscript ID: RSPB-2020-2526

“The under-investigated wild side of *Escherichia coli*: genetic diversity, pathogenicity, and antimicrobial resistance in wild animal hosts”

by Katherine Lagerstrom and Elizabeth A. Hadly
Stanford University

We appreciate the constructive comments provided to us by the referees and have worked to address all comments in a way which we believe has improved the manuscript. Importantly, a paragraph describing the workflow of our literature review was added at the beginning of the **“KNOWLEDGE GAPS IN THE LITERATURE”** section. In response to suggestions made by both referees, we have enhanced the review’s emphasis on livestock-wildlife transmission of *E. coli* by breaking the section **“Potential routes of transmission”** into subsections and adding one entitled *‘At the livestock-wildlife interface’* with additional references, included those suggested to us. A greater emphasis has also been placed on antimicrobial resistance (AMR) by adding substantial discussion to the section **“Prevalence of antimicrobial-resistant *E. coli* in wildlife”** as well as a few additional lines on the subject in the conclusion. References have been added in each instance where a referee asked for one. Below is a detailed response to each referee’s comments. Reviewer comments are listed first followed by our responses to each in red text for distinction.

Reviewer(s)' comments to author interleaved with our response to each:

Referee: 1

Comments to the Author(s):

Manuscript RSPB-2020-2526 is an interesting and well-written review highlighting the sparse literature available to date on *E. coli* in wildlife, with an emphasis on antimicrobial resistance. I have mostly minor comments throughout. My primary concern with publication of the manuscript in Proceedings B is that many of the components are highlighted elsewhere in the literature. I believe it will make a nice contribution to the literature but isn’t a major enough advancement for Proceedings B.

My primary big comment for the authors in revising the manuscript is that I think the review should have greater emphasis on livestock-wildlife transmission. For example, Fig. 3 focuses on an axis of increasing human population density and urban impact. However, exchange of *E. coli* (and other foodborne pathogens) often occurs at the wildlife-livestock interface rather than urban interface and can also be the source of AMR genes. Michele Jay-Russell’s lab has quite a lot of literature on the matter, but to give some specific examples of a few refs of many that show or suggest wildlife-livestock transmission of foodborne pathogens: Carlson et al. (2015) <https://doi.org/10.1016/j.vetmic.2015.04.009> (although this is Salmonella), Smith et al. (2020) <https://doi.org/10.1111/1365-2664.13723>, Hald et al. (2016) <https://link.springer.com/article/10.1186/s13028-016-0192-9> (Campylobacter but suggests bird/livestock transmission), and Rivadeneira et al. (2016) <https://escholarship.org/uc/item/3733r6pf>. In contrast, the studies that have tried to examine the impacts of urbanization on foodborne pathogen prevalence in birds have largely failed to see an effect, e.g., % urban didn't matter in Smith et al. (2020) <https://doi.org/10.1111/1365->

2664.13723, Rouffaer et al. (2016) (<https://journals.plos.org/plosone/article?id=10.1371/journal.pone.0155366>), Brobey, Kucknoor & Armacost (2017) (<https://doi.org/10.1637/11607-020617-RegR>), nor Hamer, Lehrer & Magle (2012) (<https://doi.org/10.1111/j.1863-2378.2012.01462.x>) (but Hernandez et al. (2016) did see an impact of urbanization on white ibis; <https://doi.org/10.1371/journal.pone.0164402>).

The section titled ‘Potential routes of transmission’ has been divided into sub-sections and one was added specifically focusing on livestock-wildlife transmission of *E. coli*. Figure 3 has been amended to emphasize the impact of livestock on AMR prevalence in wildlife-associated *E. coli* by changing the axis in (c) to illustrate a natural to farm/livestock environment rather than a natural to urban gradient, as suggested.

I like the title.

Abstract:

L16-18: There are not too many cases where wildlife were definitively linked to outbreaks of *E. coli*. I would say, “...are sometimes implicated as the source of pathogenic...” or “have been implicated.” The primary ones that I am aware of were the 2006 spinach outbreak in California in which they found matching isolates between pigs and cattle near the field, but the final CDC report said water flooding could have been responsible. There was also an Oregon outbreak thought to be caused by deer. (These are mentioned at some point in the article). In any case, I think the language is a bit strong at this sentence.

The wording in L16-18 has been changed in the way suggested, removing the word “often” to say “have been implicated.”

Introduction:

There are no citations from lines 32-46 to support the statements made. The statements themselves seem accurate, but the authors should add citations.

Citations have been added throughout the introductory paragraph referenced here.

L49-51: The authors never specified any detail on how studies were acquired. Was there a systematic search? What were search terms used? Did they search literature that came up in their search for more references? How can the authors be sure their protocol sufficiently acquired studies? For example, one citation missing in references that is very relevant is Navarro-Gonzalez et al. (2020) <https://aem.asm.org/content/86/3/e01678-19.abstract>

A detailed description of our literature review methods has been added to the beginning of the “Knowledge Gaps in the Literature” section (L78-L111). All relevant papers were assembled through a series of keyword searches in Google Scholar followed by more careful handpicking by reading through abstracts and methods, as well as by reviewing the literature obtained in our keyword searches for any additional references that were missed by the keywords chosen. The citation which was noted as missed (Navarro-Gonzalez et al. (2020) <https://aem.asm.org/content/86/3/e01678-19.abstract>) has also been added to our review in the relevant livestock-wildlife interface subsection on potential routes of transmission, though it should be noted that this reference studied foodborne pathogens in wild birds in California, and a single sequence type of *E. coli* (*E. coli* O157:H7, and a very small percentage of non-O157 STEC) was one of many diverse pathogens considered in this paper and therefore it is only minorly relevant to the intended scope of this review which was primarily to focus on *E. coli*

genetic diversity in wild animals. We acknowledge that we likely missed other references that only refer to this single sequence type of *E. coli* and its role as a foodborne pathogen. This was intentional, as *E. coli* O157:H7 is one of the most studied pathogenic strains of *E. coli* for its relevance to human health and therefore including every single article that mentioned *E. coli* O157:H7 would have distracted from other elements within the scope of this review (e.g. ABR, genetic diversity, transmission within and between wild and human/livestock populations).

L51-53: Here, I will explain why I do not think the findings are novel enough for Proceedings B. I think the review is very valuable but would fit better at a discipline-specific journal. For example, Smith et al. (2020) <https://onlinelibrary.wiley.com/doi/full/10.1111/brv.12581> outlined points 1 and 3 (inconsistency between studies and bias and underrepresentation of wild bird hosts studied). The geographic bias outlined in the paper is an issue in the broader literature across topics of investigation, e.g., Clarke et al. (2017) <https://doi.org/10.1016/j.tree.2017.02.012> Greater emphasis on the AMR and genetic diversity aspects of the review, to me, would stress the novel aspects because I do not believe they are well-covered for *E. coli* in a review.

Throughout our revisions and in addressing both referees comments, we strove to place even greater emphasis on the presence of AMR in wildlife and its implications for global public health and wildlife management. We also emphasized the importance of investigating the genetic diversity of *E. coli* in wildlife in order to better inform our understanding of the ecology of the bacterial species as a whole, which will in turn better equip us to monitor potential threats of AMR *E. coli* spillover and transmission between wild and human populations, as well as to trace and thus prevent agricultural contamination by pathogenic *E. coli*. We believe the geographic biases in coverage identified in our literature review are significant and unique to this review, as such a collection of papers has never been assembled before. Geographic biases in where *E. coli* has been studied in wild animal populations poses issues unique to the topic because it disallows answering questions important to protecting global public health such as; is the distribution of *E. coli* geographically-determined, host specific or ubiquitous? Are there higher rates of AMR in countries with weaker controls on antibiotic use in both medical and agricultural sectors? Are rates of clinical cases of AMR pathogenic *E. coli* higher in countries where people have more contact with wild animals?

L72-88: I think the results emphasized at L72-88 could go in an appendix in favor of highlighting information on wildlife-livestock transmission and methods to acquire literature. The information presented has largely been covered elsewhere.

The results outlined at L72-88 are new data specific to and uncovered by our review of the literature published to date on *E. coli* in wild animals and have not been published elsewhere. The section was left in place, too, because referee 2 suggested additional discussion here regarding other explanations for the lack of quality data on the topic globally. However, wildlife-livestock transmission has been more strongly emphasized in our revision, as we have added a new section on the wildlife-livestock interface which we referenced in response to this referee's first comment.

L81-88: This result has previously been identified by Smith et al. (2020) Biol Rev for birds. I am unsure if anyone has robustly summarized this for mammals.

To our knowledge, this is the first time this result has been robustly summarized for mammals.

Section starting at L113: I think this section is a more novel component to review. I would

emphasize these results. I'd also suggest mentioning more of the role of antibiotic use in livestock in the carriage of AMR genes by birds.

As we have noted in previous responses to comments, we have made revisions to emphasize the occurrence and implications of AMR *E. coli* in wild animal populations as well as in livestock and at the interface between livestock and wildlife. This includes the addition of a reference in our section entitled "At the livestock-wildlife interface" citing the identification of STEC on the feet or feathers of two wild birds in CA as well as shared-strains in wild geese and free-range cattle, suggesting a common source of contamination in the environment and potential transmission between species.

L125: I would add a reference to the WHO report page

A reference to the WHO webpage has been added here (now L251).

L143-144: How prevalent?

This remains an open question in the field. It is difficult to estimate due to the fact that very little research effort has been made thus far to quantify the levels of ESBL-*E. coli* present in diverse wildlife populations. Language has been added here to clarify this and suggest the need for further investigation.

L176: Add references to support statements

Multiple references have been added to support the statements made in L176, now L348-L356.

L195: I am only somewhat familiar with sequence databases, but I am not entirely sure this is true. There is PubMLST <https://pubmlst.org/organisms> (seems more promising) and PulseNet <https://www.cdc.gov/pulsenet/index.html>

Clarifying language was added here to support the statements made, for a "comprehensive and universal database"... "when wildlife are involved." The database cited here as being more promising (PubMLST) was accessed and searched for *Escherichia coli* and it was found that the majority of isolates in the database (of only 279 total) were from food sources, and also clinical and environmental sources. A few were listed as sourced from "animal" but for the majority, that was as specific as the source classification got, leading us to believe it came from a domesticated animal or livestock, as most isolates studied to date are. (A couple did specify "pig faeces" which is likely not wild, but livestock.) This further supports our call for deeper investigation into wild animal *E. coli* and the need for the creation of a comprehensive database of *E. coli* derived from wild animals, because very little information exists to date.

L199: Add reference to support that wildlife are implicated as sources of contamination

Two sources were cited here; Karp DS, Gennet S, Kilonzo C, Partyka M, Chaumont N, Atwill ER, Kremen C. 2015 Comanaging fresh produce for nature conservation and food safety. *Proc. Natl. Acad. Sci.* **112**, 11126–11131. (doi:10.1073/pnas.1508435112) and Beretti M, Stuart D. 2008 Food safety and environmental quality impose conflicting demands on Central Coast growers. *Calif. Agric.* **62**, 68–73. (doi:10.3733/ca.v062n02p68)

L204-206: Generally, livestock, worker sanitation, and water are considered bigger risk factors for food safety, with wildlife considered a pretty minor contributor. E.g., Parker et al. (2012)

<https://link.springer.com/article/10.1007/s10460-012-9360-3> or Park et al. (2013)
<https://aem.asm.org/content/79/14/4347.short>

This observation could simply be due to the fact that wildlife-harbored *E.coli* strains are chronically understudied. There are many cases where agricultural contamination by *E. coli* occurs and a source of that contamination is never identified, even though efforts are made to do so. One likely but never confirmed zoonotic case of pathogenic *E. coli* transmission to humans was cited in our review, at the San Diego County Fair, but many others exist, for example: <https://www.foodsafetynews.com/2020/10/mystery-surrounds-two-new-e-coli-outbreaks-with-genetic-links-to-past-romaine-events/>.

L207: “birds and insectivores” -> “insectivorous birds and bats” or something similar. Also, add reference to support benefit of ecosystem services.

This was changed to insectivorous birds and bats and we have clarified what was meant by the reference to ecosystem services. Those services referred to encompass all that we know (and don't know) that players in an ecosystem provide us, such as what native species do to support the ecosystem and food chain (population and pest control, pollination etc).

L217-218: I would say this is an unfair assessment. There are protocols in place, but it is challenging.

The language here was adjusted to say “The protocols in place to respond to such outbreaks are inconsistent and challenging to implement with current tools and understanding of pathogenic *E. coli* transmission.”

L221: Reference? The Park et al. article mentioned above would work here.

The Park et al. (2013) <https://aem.asm.org/content/79/14/4347.short> reference was cited here.

L246-249: I suggest breaking this into two sentences. I believe “effecting” should be “affecting” or “impacting”

Changed to “impacting”

Conclusion:

L260-264: I think that comparing COVID spread to *E. coli* spread is unrealistic because *E. coli* is a fecal-oral pathogen whereas COVID is respiratory. According to Batz et al. (2012), <https://doi.org/10.4315/0362-028X.JFP-11-418>, there are ~65,153 cases of O157:H7 and 112,752 cases of non-O157 in the US each year. According to the WHO, there are 8,403,121 confirmed cases of COVID in the US at this time (a partial year). Therefore, I think due to the very different transmission pathways, *E. coli* is unlikely to be comparable to COVID, even with AMR genes.

We removed the reference to COVID-19 in the conclusion (per referee 2's comment as well) because the points they have made are fair and because much has evolved with the pandemic since the first submission of this manuscript, yielding the comparison unfair.

L279: I believe “effect” here should also be “affect”

We are referring to the resulting changes made rather than the forces making them, so effect is correct (Word tags “affect” as incorrect grammar, as well).

Supporting information: I would prefer one merged file instead of having to download 4

The 4 supplementary files have been merged to a single document.

Referee: 2

Comments to the Author(s)
Review RSPB-2020-2526

General comments

This manuscript is a review of the (rare; less than 100 articles) existing literature on *E. coli* studies in wildlife populations in situ. It provides information about the geographic and host distribution, the AMR found and its pathogenicity. The authors rightfully highlight that there is little information available on this topic, that there is a risk for global public health and that the research community should focus on this compartment. Finally, the authors conclude by presenting some research priorities.

The manuscript is timely, of interest and well-written. I have a few comments that could improve its quality.

Firstly, I was expecting to see a summary of the main phylogroups found in wild hosts compared to what is found in other domestic and human hosts. There is no comparative approach for non-specialists. For example, I would have liked to know if wildlife hosts have a “phylogroup profile” closer to livestock (predominance of B1)? Or to human? Or if this phylogroup profile was more impacted by geography? By animal order (mammals vs. birds)? By latitude as biodiversity? I guess one cannot answer those questions with the current data, but these would be interesting questions to raise in the manuscript.

Here we added a brief summary to the “Genetic diversity of *E. coli* in wild animals” section describing the ecological roles of the different *E. coli* phylogroups that have come to light in the literature so far. We also added a line to conclude this paragraph expanding further on the open questions in the field outlined here.

Secondly, regarding AMR *E. coli* in wildlife: there is not a clear section presenting AMR as being the product of a natural process and (pre)existing in nature. Today we have the tools to differentiate between natural and anthropogenic AMR strains. What we observe in the few existing studies is what you describe as AMR being mainly anthropogenic and percolating into the wild. Why did you not take that perspective? (you still mention their “prevalence in natural environments” L136). In addition, the evolution of AMR genes in the wild merits some comments: a priori they should be counter-selected in environments with low AB pressure but some studies indicated that there could be AMR genes with no cost to its host.

We added language and an additional reference (Martinez JL. 2008 Antibiotics and antibiotic resistance genes in natural environments. *Science* (80-.). **321**, 365–367. (doi:10.1126/science.1159483)) to acknowledge the environmental origin of antibiotics and the associated resistance mechanisms in natural bacterial communities. Comment was made on the evolution of antibiotics and AMR in natural environments and additional supporting research

was cited to argue for the anthropogenic sources of AMR in wildlife. We also addressed the “counter-selection” argument by citing a study (Klümper U, Recker M, Zhang L, Yin X, Zhang T, Buckling A, Gaze WH. 2019 Selection for antimicrobial resistance is reduced when embedded in a natural microbial community. *ISME J.* **13**, 2927–2937. (doi:10.1038/s41396-019-0483-z)) which addresses this question and found that the level of resistance does decrease in the absence of AB pressure and that resistance genes often do confer a cost to the organism carrying them, though there is more work to be done to understand the cost of harboring ABR genes in a diversity of bacterial species. It is likely a different cost for different species and in different genetic backgrounds and environments.

Thirdly, I would like the authors to question the relative risk of AMR in wildlife hotspots on public health. Of course, there is a risk. But in a resource-limited research context, is this research a priority? I would expect the authors to discuss the issue in the broader AMR context and global public health. For example, from a global public health perspective, who are the humans most impacted by AMR principally (us farmers and consumers vs hospital patients)? Where are they living? Do we know if they are greatly exposed to AMR from wildlife? This question is important regarding the global fight against AMR. Do we really need to invest into AMR in wildlife research or we should prioritize other AMR research sectors?

This point was also addressed in this referee’s specific comment on L24-25 (Abstract, below). We are missing a big piece of the puzzle in our global efforts to combat AMR if we ignore AMR in wildlife. All humans referenced here; farmers, consumers and hospital patients, are at risk of contracting an AMR infection derived from a wild host reservoir, some more directly than others, but because *E. coli* transmits readily via the fecal-oral route, we are all potential recipients. That said, due to the current severe lack of understanding about the prevalence and role of AMR in wild animal populations around the world, sound predictions of the level of threat and its varied impacts on different sectors of our society, remain largely unknown. It is difficult or impossible yet to answer the questions posed here, that is why the area needs more research. Let us point out too, that AMR has been reviewed multiple times in wild animals to date and research in the field is already expanding. AMR is just one topic we want to address when considering the ecology and evolution of *E. coli* in wild animal hosts. Our primary focus is on the bacterial species in wildlife, the presence of AMR in wild animal microbiomes at large is just another piece of the picture of the role of *E. coli* genetic diversity in wild animal hosts. We would love to spend more time dissecting the scope of AMR in wildlife and the priority of its research for public health, but it is beyond the scope of this review.

Then, your article is written mainly with the perspective of the impact on global public health. Dealing with wild hosts, you could also in the introduction and maybe in the conclusion discuss potential biodiversity conservation impacts at hotspots level but also at genetic diversity level? Same remark for the impact of AMR from and on livestock production?

We added a section about AMR and *E. coli* transmission at the livestock-wildlife interface, mentioned previously. For the sake of brevity, we determined that a discussion of the impact of AMR generally on livestock production was beyond the scope of this review. There is indeed much research being conducted in this arena, but keeping with our efforts to summarize the current state of knowledge about *E. coli* in wild animals, we chose not to get too deep into the literature on livestock AMR prevalence. However, we did amend the section title “Gradients of wildlife-human contact and AMR prevalence” to include livestock (“Gradients of wildlife-human/livestock contact and AMR prevalence”), because the reviewer makes an important point, that livestock play a large role in AMR dissemination, perhaps even more so than strictly human/urban environments themselves, a point also made by referee 1.

The impact of *E. coli* genetic diversity and the presence of AMR and pathogenic *E. coli*-associated virulence factors in the gut microbiomes on the biodiversity and conservation of wild hosts is largely unknown. Stress, such as that resulting from factors that negatively impact biodiversity like habitat destruction and climate change, is known to cause shifts in the microbiome of wild animals toward lower diversity and higher pathogen load which would lead to higher rates of pathogen/AMR shed into the environment and thus transmission among wild individuals and therefore potentially into human populations. It is still unclear whether or not harboring pathogenic (to humans) strains of *E. coli* causes an immune response within wild individuals or if the presence of AMR in wild microbiomes negatively impacts biodiversity and therefore conservation efforts. Comment on this point was made in the introduction as suggested.

Finally, I was expecting a few lines on the difficulty and requirements linked to sampling *E. coli* in wildlife hosts, especially in understudied countries where sampling, storing and sending (practically and legally) samples from wildlife could be difficult. This can also explain the paucity and the lack of quality of data available and the need to collect relevant meta-data when working with wildlife (e.g. species, size of group, estimating age, estimation of the wildlife/livestock/human interface and interactions with other wild species etc.).

A few lines were added to address the difficulty and requirements of sampling *E. coli* in wildlife hosts in the section where understudied countries were identified (L126-137). Other possible explanations for the lack of quality of data available were outlined, including barriers to obtaining legal permissions and collecting all the necessary and relevant meta-data when working with wildlife in the field.

Specific comments

- There is no information about how the literature review was implemented. Could you provide briefly the process? For example, which keywords did you choose to find the <100 wild animal studies out of the > 2 million *E. coli* studies?

We added a paragraph to the “Knowledge Gaps in the Literature” section detailing how the literature review was conducted and what keywords were used (L78-L111).

- In addition you mention in the abstract the “mammalian gut microbiome”, in the title “wild animal”, then you talk about mammals, birds and in Figure 4 there is also a fly. Did you limit your review to some orders? (no mention of reptiles because there is no study on reptiles or because you did not review them?). The reader would need clarification on this.

We clarified the scope of the review (we did not exclude reptiles, it is just the case that very few studies exist on reptilian-associated *E. coli*) within the description of how the literature was reviewed.

- L24-L25: Abstract: “because of the strong implications for global public health”: are you confident that the role of “wildlife” is relatively that important? Do we have proofs for that? (I mean a significant amount of proof or we don’t know the risk but it could be important?). That’s the main argument that was said to me when I tried to justify for funding for AMR in wildlife (and did not get it!).

Zoonotic disease emergence is a rising problem we will increasingly face in the Anthropocene as human populations continue to expand and invade previously wild lands and as humans come into contact with more wildlife through this expansion and resource extraction. Additionally, global climate change will allow more pathogens to persist in warmer climates, or emerge from thawing climates. This is evident by the likely source of the current pandemic, thought to have originated from a wild bat. The increasing presence of AMR in human clinical cases of infection as well as in wild animal populations only further confounds this problem, as it will become increasingly more challenging to combat these diseases as our previously relied on drug treatments become ineffective. It is a bit subjective when it comes to defining a threshold beyond which “we” can consider the impact of AMR in wildlife a significant enough threat to global public health to justify research funding. The level of severity or concern that defines that threshold will be different depending on with whom you talk. But the goal of this paper is to increase awareness of this impending problem and convince more people that we need to act, and fund that research, now. A few words were added after the statement “because of the strong implications for global public health” in the abstract to emphasize this goal.

- L37-38: “identifying sources of agricultural contamination of pathogenic *E. coli*”: specify “natural” or “wild” *E. coli* because at first reading I thought you were talking about contamination from agricultural origin into the wild.

We reordered the words for clarity; “identifying the source of pathogenic *E. coli* in agricultural products.”

- L46: you mention “invertebrates” but you don’t give any reference.

The reference to invertebrates was removed here, as we originally thought to cite earthworms, but because current literature only discusses *E. coli* only as it passes through or sticks to exterior of earthworms and not as it “resides” in their guts, their mention was misleading.

- L103-105: “The first study to examine *E. coli* in wild animals...”: I don’t understand the sentence. There is a word missing or clarify. What level of diversity are you talking about? Phylogroup I guess?

The level of diversity mentioned here was referring to the diversity index: H. Wording was added to clarify that.

- L138-140: and this leads to your concept of “melting pots”. I agree with this concept but according to your presentation the global process is: AMR is produced in anthropological systems (animal production, human populations, humanly-impacted environment), then this AMR is hosted in wildlife where it can be maintained, evolve and spill back to human or production animals. In your melting pot, “natural” AMR does not seem to play a role (and it is reinforced by my point above). Don’t you think that the mixing of anthropogenic and natural AMR is a potential threat for “novelty”? If yes, maybe you should have it clearly said in this section. Do we have any indication of a pathogenic strain that have evolved in wildlife, spilled back into humans or livestock with a higher pathogenicity? If no, and if I agree that this is a risk, you should relativize this risk.

As referee 1 points out, “There are not too many cases where wildlife were definitively linked to outbreaks of *E. coli*.” We have so far identified very few instances in which *E. coli* certainly transmitted from a wild animal to a human. Because of this, we are unaware of any case wherein a pathogenic strain was identified in humans, then in wildlife, and then back in humans

or livestock. This would be very difficult to trace considering the extent of genetic evolution that would take place in the process. How would we show that the “new” pathogen is indeed a descendant of the original pathogenic strain and not a closely related, but differently sourced strain? This would require deep sampling of multiple hosts (wild, human and livestock) as well as WGS of numerous *E. coli* isolates, screened over many years. We know transmission is likely to occur between wild and human populations by looking at the sequence types (STs) of *E. coli* present in both populations. Time and again, research has shown that wild animals and humans possess *E. coli* of identical STs, but to our knowledge, research has yet to identify a specific pathogenic strain that has evolved in wildlife and spilled back into humans or livestock, with higher pathogenicity. The lack of such an identification can be related to the difficulty discussed with tracking strain movement. Additionally, a big factor making the identification of this occurrence difficult is the fact that pathogenic strains of *E. coli* are by nature *pathogenic to humans*, and therefore not defined as pathogenic until they are discovered to make a human sick. Therefore it is possible that a “new” strain evolves in a wild host, but we wouldn’t identify it as a pathogen until we see it in a human. But the referee makes a very good point here, and deeper discussion on the mixing of naturally-occurring AMR with anthropologically-derived AMR, along with the opportunity for *E. coli* and AMR to evolve in wild hosts, was added to the referenced section (now L226-L231).

- L278-286: you suggest rightly that we can “easily” follow the spread of AMR into wildlife from a human origin (because there should not be a lot of AMR in the wild); and then you say that there is already so much AMR in the wild that the risk of spill back to humans and livestock is important and we have difficulty to trace it. There is a small paradox here that you need to discuss. In this paragraph on routes of transmission, you don’t mention the wildlife/livestock/interfaces that are the places where spillover and spill-back of *E.coli* strain towards and from the wild occur. Maybe that research should also focus on some of these interfaces to better understand the ecology of *E. coli* transmission?

For the sake of clarifying the first statement made here, it is not “easy” to follow the spread of AMR into wildlife. Much more research needs to be done to understand the pathways by which AMR is conferred to wild animal-associated microbes, but it is clear that the presence of such resistance genes did not arise naturally and is a direct result of environmental contamination by humans by a diversity of means. The difficulty lies in actually tracing the spread of pathogens like *E. coli* through wild populations and into human populations, because the genetics of the bacterial species are so complicated (HGT, taxonomy, huge pangenome etc), likely to change (via mutation/adaptation), and under-investigated in wild hosts. A sentence has been added to the referenced paragraph to call for greater research efforts to investigate the transmission of AMR at wildlife-livestock interfaces because they offer significant opportunity for spill-over and spill-back of *E. coli* between domestic and wild animal populations. Research focus here will better inform the ecology of *E. coli* transmission along with its associated AMR profiles.

- L143: ESBL-E.coli: maybe explain a bit more what this strain is and that it has a human origin for the RSPB non-specialized readership.

A few sentences describing beta-lactam antibiotics, the resistance mechanisms bacteria has to them, and the global health concern imparted by them were added for the RSPB non-specialized readership.

- L210: “(...) produce production (...)”: awkward.

Changed to “agricultural production”

- L248: “effecting”?

Changed to “impacting”

- L260-261: the reference to COVID-19 is maybe a bit opportunistic and I don't think it brings a lot to the conclusion, except if you take the time to explain the challenges at the nexus agriculture/biodiversity/health and the global changes that impact our relationship with the nature, which COVID-19 is a symptom. (in the last 6 months, 100% of the articles I review mention the COVID-19 crisis and I don't think it is necessary). Maybe a reference to some integrated approaches to health would be more suitable?

We removed the reference to COVID-19 in the conclusion (per referee 1's comment as well) because the points they have made are fair and as much has evolved with the pandemic since the first submission of this manuscript.

Figure 2(a): the background “grey” does not differ much from the “light grey” countries making it difficult to locate countries with only 1 study.

The color gradient in Figure 2a was adjusted for better visibility. The light grey countries (those with only 1 study) are now light blue, while the “background” (countries with 0 studies) remains grey.

Appendix B

March 17, 2021

Response to referee and manuscript with track changes

Manuscript ID: RSPB-2021-0399

“The under-investigated wild side of *Escherichia coli*: genetic diversity, pathogenicity, and antimicrobial resistance in wild animals”

by Katherine Lagerstrom and Elizabeth A. Hadly
Stanford University

We greatly appreciate the constructive comments provided to us by the referee and have worked to address them all in a way which we believe has further improved the manuscript. Throughout the manuscript, and especially in the new sections added during the last revision process, we have reduced the length of run-on sentences as suggested. Below is a detailed response to each specific comment made by referee 1. Referee comments are listed first followed by our responses to each in red text for distinction. After all requested changes and additions were made, extensive reformatting and re-phrasing of sentences for brevity was done to cut 750 words and meet the 10-page limit. We have merged our previously entitled “*Pathogenic E. coli and the human-wildlife interface*” section under what is now “*Pathogenic E. coli in wild animals*”, as well as merged “*Gradients of wildlife-human/livestock contact and AMR prevalence*” under newly entitled “*AMR E. coli in wild animals*”. Doing so allowed us to remove some repetitive statements and improved the flow of the manuscript. Despite these changes, the content remains fundamentally unchanged. The full manuscript with track changes follows our response at the end of this document, starting on page 8.

Reviewer Comments to Author:

Referee: 1

Comments to the Author(s)

The authors did an overall good job incorporating the suggestions from the first revision. Below, I made some additional, mostly minor, suggestions. I think the article will be of interest to people studying *E. coli* and wildlife-associated food safety issues.

Generally, the paper is well-written. I suggest trying to reduce some of the really long sentences in the new sections. Many have > 50 words and are hard to follow.

We have reduced the number of words in or divided many of the overly long sentences in the newly added sections.

Abstract:

L13-14: I don't think this sentence does a lot to set up the paper, and I think it could be removed. The introductory material in the abstract takes up quite a bit of space that could be traded off to highlight findings from the review. I think it'd help the article's eventual impact since, unfortunately, a lot of people don't read past abstracts.

We've removed the first two introductory sentences and added another line highlighting the findings of our review.

L27-28: Given reviewer 2's comments, it seems more appropriate to say, "...especially in light of the potentially strong implications..."

The words "potentially strong" have been added to this sentence.

Main text:

The additional text in the introduction sets up the paper better.

L69-70: Citations to anchor the statements (that *E. coli* causes a large public health burden and ES provide high economic value) would be good.

We added citations to support this statement.

L85: "In order to" -> "to"

Changed to "to"

L117-128: If leaving this in as a major finding, I suggest adding in some of your rationale in the response to reviewers that better explains the importance of the information: "Geographic biases in where *E. coli* has been studied in wild animal populations poses issues unique to the topic because it disallows answering questions important to protecting global public health such as; is the distribution of *E. coli* geographically-determined, host specific or ubiquitous? Are there higher rates of AMR in countries with weaker controls on antibiotic use in both medical and agricultural sectors? Are rates of clinical cases of AMR pathogenic *E. coli* higher in countries where people have more contact with wild animals?" Overall, I think the current paragraph doesn't set up the rationale/importance as well as how you explained it in the response.

The section quoted here from our response has been added to this paragraph to bolster the argument of the importance of geographic biases in *E. coli* studies to date.

L154: For more general audiences, specifying what "H" is would be helpful

A brief definition of "H" has been added here.

L182-185: Is there a published/peer-reviewed example you could use here instead? An unpublished report isn't the best evidence.

Correction has been made to this sentence and a real (numerical) citation has been added. The O'Neill Report of 2014 was high profile and heavily discussed.

L194: Should this say, “antibiotics used in medicine”? I had to re-read this a few times.

Yes, the mistype has been corrected to say “used”

L195: “Piece” -> “pieces”

Corrected to “pieces”

L194-221: The sentences in this new section are pretty long. I suggest trying to break them up to be ~25 words or less each to make it easier to parse.

The sentences in this section have been revised and shortened for easier reading.

L221: I suggest being less definitive with “the risk of such an occurrence is real” since no one has documented it. I think your reply to reviewers was great as to why that is very hard to do. However, it’s pretty strong language in text. Perhaps, “the risk of such an occurrence is nonzero” or “such an occurrence could theoretically happen.” You could also rephrase the sentence to say something like, “We have not yet documented a pathogenic strain that evolved in such a way in wildlife and spilled back into humans or livestock with a higher pathogenicity. However, such approaches are highly impractical, and it would be nearly impossible to document in the wild.”

The latter suggestion was taken and the sentences “We have not yet documented a pathogenic strain that evolved in such a way in wildlife and spilled back into humans or livestock with a higher pathogenicity. However, such approaches are highly impractical, and it would be nearly impossible to document in the wild” replaced the sentence “Though we have not yet identified a pathogenic strain that evolved in such a way in wildlife and spilled back into humans or livestock with a higher pathogenicity, the risk of such an occurrence is real.”

L297-300: I think this result is a little more complex than stated. The semi-natural land cover clearing can have impacts on the wildlife species that are found on the farms. These species can vary in reservoir competence. Clearing of semi-natural land cover, may, therefore shift the community towards better hosts (e.g., Smith et al. 2020 JAE). I might suggest a change like, “Despite these environmentally-harmful measures, prevalence of pathogenic *E. coli* actually increased in many cases by more than an order of magnitude between 2007 and 2013, indicating that removal of semi-natural land cover may be ineffective (Karp et al. 2015 PNAS; Smith et al. 2020 JAE).” Then, you could emphasize at L300-303 that these practices can favor species that are better hosts (e.g., European starlings). I’d suggest a modification like, “Exclusion or elimination of native species on and surrounding farmland, such as bees, insectivorous birds and bats, via habitat destruction, trapping or killing, could potentially lead to incalculable losses of ecosystem services these wild species provide. Destruction of native land cover may also shift biological communities towards dominance by competent hosts, counterintuitively exacerbating food safety risks (Smith et al. 2020 JAE).” Then, in the

last sentence, you could mention that it's also important to understand the contribution of individual species, which has an implication for management and risk. I'd expect birds with greater degrees of synanthropy to have greater exposure/problems, for example, so farming environments that favor these species are likely riskier for public health.

<https://besjournals.onlinelibrary.wiley.com/doi/full/10.1111/1365-2664.13723>

<https://doi.org/10.1073/pnas.1508435112>

All suggested modifications outlined here have been made to this section and a few sentences were added second to the end of this paragraph pointing out the importance of understanding the fact that different species will pose different levels of risk to public health in this arena.

L313-314: I might say, "an important gap in monitoring of foodborne pathogen transmission" or "an important gap in monitoring of enteric pathogen transmission" since they thought the cases were due to petting animals rather than consuming food.

This line has been adjusted to say, "The inability to trace this outbreak to a source points to an important gap in monitoring of foodborne pathogen transmission."

L319: "In order to" -> "to"

Changed to "to"

L321-322: There are a few references that discuss these issues, and adding a citation to anchor the statement would be good. Already cited in the manuscript was the Beretti and Stuart example, but Danny Karp and colleagues have also published a bit on this tension. A recent and in depth one: <https://link.springer.com/article/10.1007/s10460-020-10123-8>

Citations were added to anchor this statement.

L344: The paragraph as edited might be more accurately subtitled, "Carried by birds" or "Movement by birds"

The word "migratory" has been removed from this header, which now states "Carried by birds". We will note here also that we have changed the order of these sections so that they are now in order of significance with respect to the areas to consider transmission. The order is now; livestock, then birds, then via social interactions among wild species.

L355: I assume you'll need the scientific name for specific species mentioned like Glaucous-winged gull

The scientific name has been added for Glaucous-winged gull (*Larus glaucescens*) L318, as well as Egyptian vultures (*Neophron percnopterus*) L200, wild baboons (*Papio cynocephalus*) L194, Rocky Mountain elk (*Cervus elaphus nelsoni*) L239, mountain brushtail possum (*Trichosurus vulpecula*) L341, and desert warthog (*Phacochoerus*

aethiopicus), common duiker (*Sylvicapra grimmia*) and African buffalo (*Syncerus caffer*) L375.

L361: "It was found that" reads a little awkwardly. Perhaps revise to something like, "Rivadeneira et al. found that wild birds can carry STEC between CAFOs (concentrated animal feeding operations) and leafy green fields; about 5% of birds in their study carried STEC (74)"

The suggested change has been made as written here.

At 365, rather than "they identified," I suggest, "LeJeune et al. (75) identified"

"they identified" has been changed to "LeJeune et al. (75) identified"

Conclusion

L:415-421: I think it is of note to also mention it's important to sample a wide-range of species to tie back into conservation and ecosystem services that is brought up throughout. Are species of conservation concern frequently harboring/spreading virulence and AMR genes or do they rarely harbor/spread them? If species that are key transmitters differ from those that are of conservation concern and those that provide services, then that suggests managing for disease, conservation, and ecosystem services aren't at odds.

"From a breadth" of wild animal species has been added at L361 to tie back this point made earlier in the article. We added a line to address the question of whether "species that are key transmitters differ from those that are of conservation concern".

We have also added to the first paragraph of the conclusion more detailed statements regarding the threat to global public health potentially posed by wild animal-harbored *E. coli* to further tie back to recurring themes in the rest of the review that weren't explicitly stated in the last draft of our conclusion.

L432: I'd spell out Democratic Republic of the Congo as not all readers may pick up on what the DRC is

DRC has been changed to "Democratic Republic of the Congo"

L432: "in order to" -> "to"

Changed to "to"

Figures:

Fig. 2: "Countries in grey were not represented in this literature review" makes it sound like you didn't include them on purpose. Perhaps say, "No studies fitting our inclusion criteria were found in countries in grey." Also, the numbers in Panel B are hard to read/see. It might work better if you put the numbers in the white space with a line going

to the country.

The sentence about the grey countries has been changed to the one suggested and the numbers in Panel B have been moved to the white space with arrows going to each country.

Table S1. “References follow the table” would be clearer.

The description for Table S1 has been changed accordingly.

Table S3. It would be good to check over the nomenclature and consistency in what words are capitalized, etc. on Table S3. It would be helpful to indicate what you followed for your common and scientific names and the date you searched for taxa since things change rapidly and differ between, e.g., IUCN and American Ornithological Society. For example, you use “common starling” over “European starling.” I think whatever you used is fine, but it would be good to know who you are following. I suggest italicizing the scientific names (assuming this is journal policy).

We have gone back through the table to check nomenclature and made consistent what words are capitalized (only proper nouns). All scientific names have been italicized and we have done our best to standardize common species names throughout the table.

Also, it is an interesting table but would be more informative/impactful if you had a label for if it is a bird or mammal. It would also be more useful to readers if you added the references that include the species to make it easy to locate individual ones, if say, you wanted to learn more about rock pigeons.

We have added a column “Class” as suggested here to denote bird/mammal/fish/reptile. We also considered adding references to the table here upon its initial creation, but opted against doing so for a few reasons. One being that many species were studied in multiple references (some up to 13) and including them all would look messy in a table format. Additionally, many studies investigated multiple species, dozens, one even looked at 166 species, so the same reference would have to be listed 166 times. So, this didn’t seem like the best format to convey the reference information. At least there are only 93 studies referenced in total, so it wouldn’t be all that hard to find your species of interest.

Lastly, upon reading the table, I noticed some specific typos or issues, but I suggest carefully going back over it for others.

Table S3 has been spell-checked in full

Specific things I found: I’d make “magpie” “Eurasian magpie” since there are many magpie species.

“Eurasian” has been added to the specific magpie row (*Pica pica*)

Calling *Aegypius monachus* “black vulture” over Cinereous vulture is also confusing since *Coragyps atratus* is the “black vulture,” according to the American Ornithological Society official common names (and American black vulture in IUCN). *Coragyps atratus* is also listed separately in the table. I think the original reference used “black vulture,” but I’d try and standardize to IUCN or something similar.

The common name listed for *Aegypius monachus* has been changed to “cinereous vulture” and the common name for *Coragyps atratus* has been amended to “American black vulture”

House sparrow should be “*Passer domesticus*.”

This correction in species name has been made for house sparrow

American redstart is missing the “t” on “redstart.”

A “t” has been added to “redstart”

“Cotte duck” is missing a scientific name.

We’ve asked a number of wildlife/biodiversity experts what they know about a “Cotte duck” and no one was able to provide a scientific name. Ahmed et al. 2019 mentioned sampling “20 Cotte duck” but did not provide a scientific name themselves. We have inserted “(scientific name unknown)” at this location in the table for consistency.

“Hoepoe” is a “hoopoe.” This appears to be Eurasian hoopoe specifically.

“hoepoe” has been corrected to “hoopoe” and re-placed by alphabetical order

Haliaeetus albicilla is labeled “sea eagles,” which seems to be referencing a genus *Haliaeetus*, but then white-tailed eagle appears to be the specific eagle mentioned.

“sea eagles” has been changed to “white-tailed eagle” and re-placed by alphabetical order

“Pygmy” is misspelled in “western pygmy possum.”

This misspelling has been corrected; pmgmy -> pygmy